# AMORTIZED NETWORK INTERVENTION TO STEER EXCITATORY POINT PROCESSES

**Zitao Song, Wendi Ren & Shuang Li** [*]
The Chinese University of Hong Kong, Shenzhen
{zitaosong,wendiren}@link.cuhk.edu.cn, lishuang@cuhk.edu.cn

## ABSTRACT

Excitatory point processes (i.e., event flows) occurring over dynamic graphs (i.e., evolving topologies) provide a fine-grained model to capture how discrete events may spread over time and space. How to effectively steer the event flows by modifying the dynamic graph structures presents an interesting problem, motivated by curbing the spread of infectious diseases through strategically locking down cities to mitigating traffic congestion via traffic light optimization. To address the intricacies of planning and overcome the high dimensionality inherent to such decision-making problems, we design an Amortized Network Interventions (ANI) framework, allowing for the *pooling* of optimal policies from history and other contexts while ensuring a *permutation equivalent* property. This property enables efficient knowledge transfer and sharing across diverse contexts. Each task is solved by an H-step lookahead model-based reinforcement learning, where neural ODEs are introduced to model the dynamics of the excitatory point processes. Instead of simulating rollouts from the dynamics model, we derive an analytical mean-field approximation for the event flows given the dynamics, making the online planning more efficiently solvable. We empirically illustrate that this ANI approach substantially enhances policy learning for unseen dynamics and exhibits promising outcomes in steering event flows through network intervention using synthetic and real COVID datasets.

## 1 INTRODUCTION

Discrete events, such as the spread of infectious disease or instances of traffic congestion, can be modeled as excitatory temporal point processes over dynamic networks. Guiding the event flows by adjusting the network structures is driven by many societal and environmental problems. For example, in the face of widespread epidemic outbreaks, governments may choose to take actions such as temporarily locking down major cities or imposing travel restrictions, with the aim to curb the spread of diseases via cutting off some edges in the disease transmission networks (Salathé & Jones, 2010; Sambaturu et al., 2020). Similarly, a common way to alleviate traffic congestion is to optimize traffic light schedules, with the hope of controlling traffic congestion flows by adaptively altering the dynamic traffic networks.

Effectively tackling these problems at a large scale is challenging due to a usually huge action space. To curb the spread of epidemic diseases, decision-makers may have to decide whether to suspend each flight across the country for each week. Similarly, to optimize traffic light schedules, city planners need to determine the appropriate actions for each traffic light within the city's infrastructure at each decision time. Moreover, different regions might face unique constraints. For example, when choosing to suspend flights to control infectious diseases, governments have to balance health concerns with economic impacts and public opinions, which can vary from one region to another. Systematically solving this network intervention problem at scale requires innovative solutions.

In this paper, we adopt a divide-and-conquer strategy and propose dividing the original policy learning problem into subproblems by splitting the original dynamic graph into smaller regions (subgraphs). In this way, the policy learning becomes flexible and manageable for each subproblem, and all these subproblems will be solved in a sequential and alternating way.

---

[*]Corresponding author

Specifically, we propose an Amortized Network Interventions (ANI) framework, which aims to learn an amortized policy shared by all subproblems via learning invariant representations of optimal policies across subgraphs with diverse dynamics. We aim to pool the policy learning from different regions and adapt to unseen new regions. Moreover, the representations for the policies should preserve the permutation equivalent properties, which implies that the network intervention policy should adapt the ordering of actions to the ordering of nodes within a graph. We achieve this by designing a bi-contrastive loss function. The permutation equivalent properties encourage encoding the most intrinsic policy information to the representation and enable the adoption of similar policy structures in analogous temporal dynamic systems.

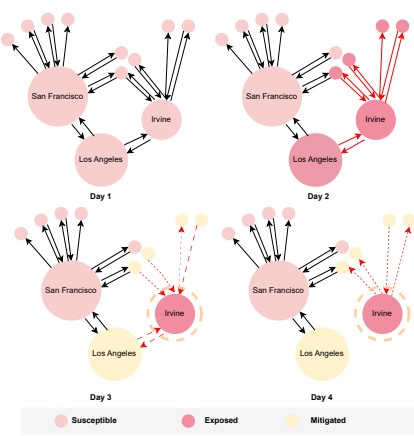

Figure 1: An illustration depicting the spread of a viral infection, where cities are represented as nodes and roads as edges.

Within each subgraph (task), the optimal policy is obtained via an H-step lookahead model-based reinforcement learning (RL), which provides a versatile formulation that can quickly adapt to changing circumstances and accommodate various local constraints. Classic infectious disease models SIR (Weiss, 2013) or traffic density model LWR (Lighthill & Whitham, 1955; Richards, 1956) are based on ODEs or PDEs. These physics-informed models offer simplified yet insightful representations of disease dynamics or traffic patterns. Inspired by these models, we use a flexible Neural ODE (Chen et al., 2018) to model the dynamics of the networked excitatory point processes. When we perform the online planning, instead of simulating rollouts from the learned Neural ODE model, we further derive an analytical mean-field approximation for the event flows given the dynamics. This approximation enables efficient policy learning for each subproblem with less variance.

In summary, we propose a systematic and deployable ANI framework that addresses the challenges of steering event flows within large dynamic graphs. Our main contributions are:

- We learn a shared amortized policy cross subgraphs to pool the optimal policy results from other regions and enable quick adaption to new regions.

- We design a bi-contrastive loss function to ensure the policy representation has a permutation equivalent property.

- We utilize a flexible H-step lookahead model-based RL approach within each subgraph and derive an analytical mean-field approximation for event flows given neural ODE models, significantly enhancing the efficiency of policy solving.

- We have conducted comprehensive experiments using synthetic traffic congestion data and real-world COVID datasets. The experimental results demonstrate the effectiveness of our approach in adeptly steering excitatory point processes through the control of network dynamics.

## 2 Preliminaries

Let's first consider how to obtain the discrete-time networked point processes given raw event data. For infectious disease cases, we can divide the geographical map into cells (e.g., a city), each corresponding to a node in the network. The edges are defined as the accessibility from one city to another (as shown in Fig. 1). We record new confirmed cases within a cell at each time step, creating a discrete-time networked point process. Similarly, we can use a lane-based approach to model the traffic congestion incidents as networked point processes. Each lane on the road becomes a network node. The traffic lights control the connection of the lanes. At each time step, we track the congestion count for each lane.

Mathmatically, we define a temporal network $\mathcal{G}_t = (\mathcal{V}_t, \mathcal{E}_t)$ indexed by $t = 0, 1, \ldots$, with $\mathcal{V}_t$ and $\mathcal{E}_t$ representing the node and edge sets at time $t$. The set of $N$ nodes is fixed at each time step. We observe a sequence of event spike counts for each node and obtain a spike count matrix over the graph, denoted as $\mathbf{X}_t \in \mathbb{N}^{N \times t}$, up to $t$. Here, $\mathbf{X}_t$ contains $N$ time-series of event counts, denoted as $\mathbf{X}_t = [\mathbf{x}_1^{1:t}; \ldots; \mathbf{x}_N^{1:t}]$. Adding or removing certain edges will influence the generative patterns of events. We aim to learn a policy to understand how to *steer the flow of the event counts to achieve some goal at minimal cost through sequentially adjusting the edges* $\{\mathcal{E}_t\}_{t \geq 0}$.

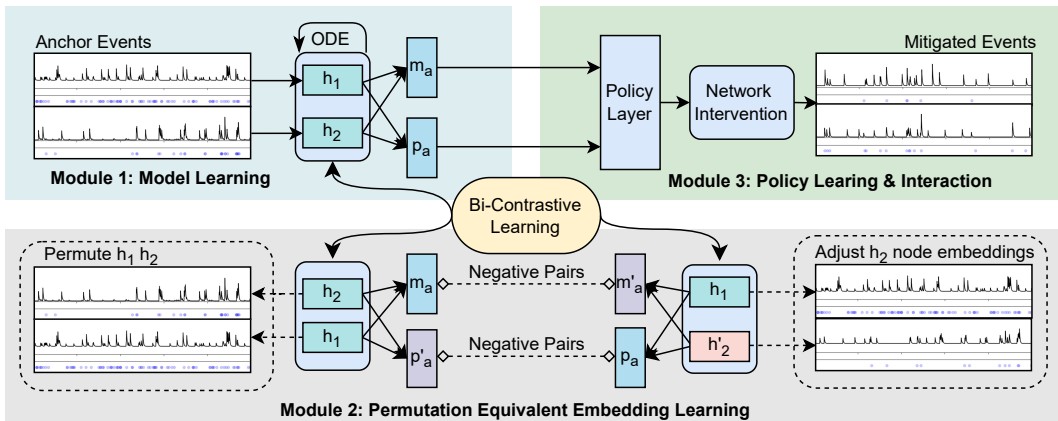

Figure 2: *Overview of the Method.* The proposed Amortized Network Intervention contains three modules. The first module is to generate a latent node embedding $\mathbf{h}_n^i$ and evolve the latent states through the NJODE model. The second module learns a Permutation Equivalent Embedding (PEE) over the latent space $\mathbf{h}_n$ by a bi-contrastive loss function prepared for the downstream adaptation. The third module accesses the learned PEE from the second module and generates a permutation equivalent policy.

We consider an infinite horizon RL problem, where an edge intervention policy, denoted as $\pi(\mathbf{h}^t)$ : $\mathcal{S} \rightarrow \mathcal{A}$, is learned to maximize the return,

$$\pi^* = \arg \max_{\pi \in \Pi} \mathbb{E} \left[ \sum_{t=0}^{\infty} \gamma^t r(\mathbf{h}^t, \mathbf{a}^t) \right], \tag{1}$$

where $\mathbf{h}^0 \sim p^0(\cdot)$, $\mathbf{h}^{t+1} \sim \mathbb{P}(\cdot | \mathbf{h}^t, \mathbf{a}^t)$, $\gamma \in [0, 1)$, and $\mathbf{a}^t \sim \pi(\mathbf{h}^t)$. Detailed descriptions are as follows: **Environment** is referring to the system that sequentially generates $\{\mathbf{X}_t\}_{t \geq 0}$ over a dynamic network $\{\mathcal{G}_t = (\mathcal{V}_t, \mathcal{E}_t)\}_{t \geq 0}$. **State** is defined as the historical event sequence and intervention trajectories up to current time $t$. We will use a compact graph embedding vector $\mathbf{h}^t$ to encode the state information (which will be discussed later), where $\mathbf{h}^t \in \mathbb{R}^{N \times D}$ and $D$ is the embedding dimension. **Action** is the choice of edge to be intervened subject to budget or other constraints. The action space is defined as $\mathcal{A} := \{\mathbf{a} \in \{0, 1\}^{N \times N} | \sum_{m,n} \mathbf{a}_{mn} \mathbf{c}_{mn} \leq B, \sum_{m,n} \mathbf{a}_{mn} \leq K\}$, where $\mathbf{c} = [\mathbf{c}_{mn}]$ is the intervention cost occurred for each edge, $B \in \mathbb{R}_+$ is the total budget at each stage, and $K$ is the maximum number of edges to be intervened at each stage. Here, we put some constraints on the action space to enable safe policies. **State Transition** dynamics are unknown for many real problems; we will build a predictive model $\mathbb{P}_\theta(\cdot | \mathbf{h}^t, \mathbf{a}^t)$ to capture the event dynamics, where model parameters $\theta$ will be estimated from observational data. **Reward Function** is tailored to suit particular applications. In general, it is a function of cumulative event counts and can be augmented by incorporating other societal or environmental considerations.

In practice, solving the RL problem (Eq. (1)) is typically computationally demanding due to its large action space and complex dynamics. This paper suggests breaking down the problem into subproblems by dividing the original dynamic graphs into subgraphs. For each subproblem, solving a model-based RL policy becomes a durable problem. Rather than solving each subproblem independently, we propose learning a shared amortized policy to consolidate policy learning outcomes across regions and transfer them to new regions. Details on the model-based RL framework for each subproblem will be discussed in the following section, with amortized policy learning deferred to Sec. 4.

## 3 MODEL-BASED REINFORCEMENT LEARNING

### 3.1 ENVIRONMENT MODEL: NEURAL ODES

Inspired by traditional ODE-based and PDE-based models in infectious disease and traffic flow studies, we propose a neural-based Networked Jumped ODE (NJODE) model to capture the event sequence dynamics. Previous works, such as Neural Spatio-Temporal Point Processes (Chen et al., 2020a) and Neural Jump SDEs (Jia & Benson, 2019), have been used to model fine-grained spatio-temporal event processes in continuous-time. We modify these models to handle large volumes of event counts in discrete-time scenarios.

**NJODE Model**: The evolving dynamics of the networked point processes are modeled by an ODE system with jumps as follows, where we introduce $\mathbf{h}_n^\tau \in \mathbb{R}^D$ as the latent state embedding,

$$\mathbf{h}_n^{\tau_0} = \mathbf{h}_n^0 \tag{2}$$

$$\frac{d\mathbf{h}_n^\tau}{d\tau} = f(\tau, \mathbf{h}_n^\tau), \quad \forall \tau \in \mathbb{R}_+ \setminus \cup_i \{\tau_i\}, \tag{3}$$

$$\lim_{\epsilon \downarrow 0} \mathbf{h}_n^{\tau_i + \epsilon} = \sum_{m \in \mathcal{N}_n} w_{m \to n} \cdot \phi(\mathbf{h}_m^{\tau_i}, \mathbf{x}_m^i). \tag{4}$$

Here, $\mathcal{N}_n$ is the neighbors of node $n$ and $\mathbf{h}_n^{\tau_i} \in \mathbb{R}^D$ is the latent state for node $n$ at time stamp $\tau_i$, where $n \in \{1, 2, \ldots, N\}$ and $\tau_i$ represents the time stamp to record discrete jumps. Rather than treating the event arrival time as a random variable (Chen et al., 2020a), we accumulate the total number of discrete events within interval $[\tau_{i-1}, \tau_i)$ and regard it as a random variable $\mathbf{x}^i$, allowing us to process large volumes of dense event data like traffic flow. The use of $\epsilon$ is to portray $\mathbf{h}_n^\tau$ as a left-continuous function with right limits at any fixed $\tau_i$. $f$ is used to model the continuous change, and $\phi$ is used to model the instantaneous jump based on neighbors' events $\mathbf{x}_m^i$. $f$ and $\phi$ are modeled as neural networks with the parameters shared for each dimension's event processes within the same subgraph. $w_{m \to n}$ indicates the influence strength from node $m$ to $n$. We denote $\mathbf{W} = [w_{m \to n}]$ as the influence matrix. This architecture is similar to a recurrent neural network with a continuous-time latent state modeled by a neural ODE. Under this formulation, the latent state $\mathbf{h}_n^\tau$ incorporates both historical information from itself and abrupt changes triggered by neighboring nodes. This mechanism for preserving abrupt change and recording memory is important to model excitatory point processes and generalize other unseen dynamics.

**Likelihood** At each time $\tau_i$, we parameterize the event count distribution as a function of the latent state $\mathbf{h}_n^{\tau_i}$. Specifically, in the rest of the paper, we assume the spike count $\mathbf{x}_n^i$ follows a Poisson distribution, whose intensity $\lambda_n^i$ is a function of $\mathbf{h}_n^{\tau_i}$:

$$\lambda_n^i = \exp(b_{\psi_n} + g_\psi(\mathbf{h}_n^{\tau_i})). \tag{5}$$

Here, we assume $g_\psi$ is the shared distribution parameter neural network among different nodes, while $b_{\psi_n}$ is the distinct baseline variable for different nodes. Given this model, we see that the final emission probability of $\mathbf{x}_n^i$ conditioned on historical observations $\mathbf{X}_{<i}$ is given by

$$\log p_\theta(\mathbf{x}_n^i | \mathbf{X}_{<i}) = -\lambda_n^i + \mathbf{x}_n^i \log \lambda_n^i - \log(\mathbf{x}_n^i!), \tag{6}$$

where $\theta$ refers to all model parameters. Finally, given a spike counts matrix $\mathbf{X}_T \in \mathbb{N}^{N \times T}$, we assume different nodes at different times are conditionally independent given the latent state $\mathbf{h}^{\tau_i}$, thereby we estimate the parameter $\theta$ by maximizing the log-likelihood that is expressed as

$$\ell(\theta; \mathbf{X}_T) = \sum_{i=1}^T \sum_{n=1}^N \log p_\theta(\mathbf{x}_n^i | \mathbf{X}_{<i}). \tag{7}$$

## 3.2 Online Planning: H-step lookahead with a learned model

Given the estimated environment model in Section 3.1, we execute online planning to learn how to perform interventions to the graph's edges to steer the discrete event flows. Specifically, for an $N$-node graph, each action involves selecting a subset of $k$ ($k \leq K$ and $K$ is the budget constraint) edges to delete from $N(N-1)$ directed edges (excluding self-connections). Hence, at the $i$-th decision time, we can represent action $\mathbf{a}^i$ as a $k$-hot matrix, resulting in the intervened influence graph given by $\mathbf{W} \odot (1 - \mathbf{a}^i)$. The state is the concatenation of the latent states for all nodes $\mathbf{h}^i = [\mathbf{h}_1^{\tau_i-}; \ldots; \mathbf{h}_N^{\tau_i-}]$.

Our approach draws inspiration from H-step lookahead online planning, which optimizes actions based on the dynamics model for a fixed H horizon with an estimated terminal value function. Only the first planned action is executed. This framework enables the prompt adjustment of the policies in real time to account for time-varying dynamic characteristics. We will adopt a policy-gradient learning algorithm and incorporate flexible constraints on the action space.

**H-step lookahead** At the $i$-th decision time, we optimize the policy $\pi$ by looking H-step ahead, i.e.,

$$\pi_{H, \hat{V}}^* = \arg\max_\pi \mathbb{E}_{\hat{\theta}} \left[ \sum_{h=0}^{H-1} \gamma^h r(\mathbf{h}^{i+h}, \pi(\mathbf{h}^{i+h})) + \gamma^H \hat{V}(\mathbf{h}^{i+H}) \right], \tag{8}$$

where $\hat{\theta}$ is the learned dynamic model and $\hat{V}(\mathbf{h}^{i+H})$ is the estimated end-of-horizon value given the terminal state $\mathbf{h}^{i+H}$. We take $\hat{V}(\mathbf{h}^{i+H}) = 0$ for H large enough, since in this scenario particular value of $\hat{V}(\mathbf{h}^{i+H})$ shouldn't affect the performance very much. Importantly, the optimal policy $\pi_{H, \hat{V}}^*$

in Eq. (8) will stabilize the nonlinear dynamic system Eq. (2-4) with the stability defined by the convergence of $\mathbf{h}^i$, as show in (Meadows & Rawlings, 1993)[Theorem 1]. During online planning, the rolling horizon is used to explore state trajectories that start from the current latent state $\mathbf{h}^i$ up to $\mathbf{h}^{i+H}$. Given new observations, we will update the dynamics and repeat the above H-step lookahead planning for the next decision time.

**Mean Field Reward** In the planning phase, the learned dynamic model $\hat{\theta}$ provides an environment simulator. Accurately approximating the expected cumulative reward demands a considerable number of rollouts. According to dynamic Eq. (4-6), obtaining each rollout involves sequentially sampling $\mathbf{x}_n^i$ and feeding them back into the neural ODEs, which must be solved numerous times. In this work, we propose to use a mean-field approximation (MFA) for $\mathbf{x}_n^i$, denoted as $\hat{\lambda}_n^i$, which will be fed back to the neural ODEs to update the latent states. The neural ODEs only need to be solved once to approximate the expected cumulative reward. Specifically, define the ground truth cumulative cost $J(t)$ at finite horizon time $t$ as $J(t) := \sum_{i=0}^t \gamma^i (\sum_{n=1}^N \mathbb{E}_{\hat{\theta}}[\mathbf{x}_n^i])$, and the mean-field estimator as $\hat{J}(t) := \sum_{i=0}^t \gamma^i (\sum_{n=1}^N \hat{\lambda}_n^i)$. The mean-field estimator uses a deterministic dynamic to approximate the averaged stochastic dynamic by averaging over the randomness in $\mathbf{x}_n^i$, allowing more efficient calculation in both forward and backward time. We theoretically justify the MFA in Appendix F. Here we provide the key result.

**Theorem 1** (**Error Bound of Mean Field Estimator**). *Let $J(t)$ and $\hat{J}(t)$ be the true and Mean Field Estimator for the cumulative cost. Let $\gamma = 1$. Suppose we have $N$ nodes. When satisfying:*

1. *The dynamic is Lipschitz, i.e., $f$ and $\phi$ are Lipschitz. And the Lipschitz constants for $f$ and $\phi$ are smaller than 1, i.e., $L_f < 1$ and $L_\phi < 1$,*
2. *The spectral radius of the influence matrix $\mathbf{W}$ is smaller than 1,*
3. *For any $n = \{1, 2, \cdots, N\}$, the intensity function $g_{\lambda_n, \mathbf{w}}$ is Lipschitz and is bounded above by $L_0$.*

*Then we have:*

$$|J(t) - \hat{J}(t)| \leq N(t+1) \cdot L_{g_{\lambda_n,\mathbf{w}}} \cdot \frac{\max_n M_n}{1 - \max_n(L_{\mathcal{T}_n})} \tag{9}$$

*where $M_n = L_{\mathcal{T}_n}(L_0^{1/2} + L_0)$ and $\mathcal{T}_n : \mathbb{R}^{N \times d} \times \mathbb{R}^N \to \mathbb{R}^d$ is the composite transition function in Proposition 1.*

**Gradient-Descent-based Optimization** Instead of exhaustively searching the discrete combinatorial action space to optimize our objective, we approximate this space using a continuous relaxation technique for the $k$-subset sampling (Xie & Ermon, 2019). We replace $\mathbf{W} \odot (1 - \mathbf{a}^i)$ with $\mathbf{W} \odot (1 - \mathbf{p}^i)$, where $\mathbf{p}^i$ represents edge selection probabilities. With this reparametrization, the objective becomes a fully deterministic function of the policy and dynamics, enabling end-to-end differentiable policy learning. We aim to solve a constrained policy learning problem. We distinguish between hard and soft constraints in our approach. For hard constraints, such as limitations on consecutive lockdown days for a county (e.g., not locking down a county for more than certain consecutive days), we employ a dynamic mask to explicitly exclude actions that fall outside the feasible space. As for soft constraints, like ensuring overall fairness in the policy, we design an additional reward term, denoted as $r_{\text{aug}}$, and scale it by some weight. This augmented reward term is jointly updated with the policy to enforce fairness within the optimization objective.

---

**Algorithm 1:** ANI (*Meta-Training Phase*)

**Input:** Task pools $\mathcal{B}$ and a pretrained model pool $\Theta$ learned based on Eq. (7)
**Result:** Policy parameters $\varphi$ and representation parameters $\psi$
Initialize parameters $\varphi$ and $\psi$;
**while** *meta-training not complete* **do**
    Sample a network $\mathcal{M}_j \sim \mathcal{B}$ and corresponding model $\theta_j \in \Theta$;
    // Policy & Representation Learning
    Optimize $\{\varphi, \psi\}$ jointly based on Eq. (8)(10);
    Obtain intervened network $\mathbf{W}'$ based on policy $\pi_\varphi$;
    // Planning ahead
    Collect new data $\mathcal{D}_j$ by *NJODESolver*($\mathbf{W}', \theta_j$) by Eq. (2 -5);
    // Adaptive Model Update
    Optimize $\{\theta_j\}$ by $\mathcal{D}_j$ based on Eq. (7) and Update $\theta_j$ in $\Theta$;
**end**

## 4 MAKING LARGE-SCALE PROBLEM TRACTABLE: AMORTIZED POLICY

In practice, managing a city's extensive traffic network is a challenging large-scale problem due to its sheer size. To tackle this, we employ a divide-and-conquer approach, breaking the problem down into manageable subproblems. For instance, we segment the vast network into smaller, more manageable subgraphs, each representing a tractable subproblem. While this strategy makes the overall problem more manageable, it raises a crucial question: How can we utilize optimal policies from previous subproblems to streamline the optimization of new ones? To address this, we introduce the Amortized Network Interventions (ANI) framework.

Our objective is to learn a shared amortized policy (Gordon et al., 2019) that can be applied across different regions with distinct dynamics. We hypothesize the existence of collective behavior among these various local temporal dynamic systems. Given a sequence of local policies $\{\pi_j\}_{j=1}^{M}$ addressing $M$ distinct sub-problems, our goal is to create an amortized policy $\pi_{ANI}$. This policy should extract invariant representations and enable the adoption of similar policy structures among similar temporal dynamic systems. The proposed amortized policy $\pi_{ANI}$ is learned over a distribution of random tasks from $\mathcal{M}$, where each task $M_j \sim \mathcal{M}$.

**Permutation Equivalent Property** Inspired from policy similarity embeddings (PSM) (Agarwal et al., 2021) and the policy permutation invariant property in SensoryNeuron (Tang & Ha, 2021), we devise an agent that can extract *permutation equivalent embeddings* and is *policy permutation equivalent* to the latent state space $\mathbf{h}^t$, where $\mathbf{h}^t = (\mathbf{h}_1^t; \ldots; \mathbf{h}_N^t)$. Since each block of $\mathbf{h}^t$ corresponds to a $N$ nodes system in the excitatory point process, the permutation equivalent property along the node dimension characterizes the global patterns while the learned node embedding $\mathbf{h}_n^t$ from Neural ODEs preserve individual's local characteristics. For detailed reference, we present the definition of permutation equivalent property and a derived Permutation Equivalent Metric (PEM) $d_\pi$ similar to $\pi$-bisimulation (Castro, 2020) in Appendix A, Definition 1 and Definition 2.

**Bi-Contrastive Metric Embeddings** We use a representation mapping $\psi$ to project the high dimensional latent graph embeddings $\mathbf{h}^t$ into two low dimensional graph embedding $\mathbf{p}^t$ and $\mathbf{m}^t$, where $\mathbf{p}^t$ only contains the internal positional information of $N$ nodes system $\{\mathbf{h}_n^t\}_{n=1}^{N}$ while $\mathbf{m}^t$ contains the magnitude information for different nodes $\mathbf{h}_n^t$. We illustrate the architecture in Figure 2. Intuitively, the graph magnitude embedding $\mathbf{m}^t$ would be invariant under row permutations of $\mathbf{h}^t$ while the graph positional embedding $\mathbf{p}^t$ would be invariant when we only change the magnitude of the row features in $\mathbf{h}^t$. During training, we perturb the anchor graph embedding $\mathbf{h}^t$ into two groups $\mathcal{G}_{\text{perm}}(\mathbf{h}^t)$ and $\mathcal{G}_{\text{mage}}(\mathbf{h}^t)$. To jointly learn the positional and magnitude embeddings with PEM, we adapt SimCLR (Chen et al., 2020b) and design a bi-contrastive learning scheme, under which the graph positional embeddings and graph magnitude embeddings can either be a positive pair under permutation transformation or a negative pair of the anchor graph under magnitude adjustment. For any anchor embedding $\mathbf{h}_0$, we take the augmentation $\mathbf{h}_1 \in \mathcal{G}_{\text{perm}}(\mathbf{h}^t)$, and $\mathbf{h}_k \in \mathcal{G}_{\text{mage}}(\mathbf{h}^t)$, $k \neq 0, 1$. Then, the bi-contrastive metric embeddings loss is given by:

$$\mathcal{L}_{BCME}(\mathbf{h}_0, \mathbf{h}_1, \{\mathbf{h}_k\}; \psi) = -\log \frac{\Gamma(\mathbf{h}_0, \mathbf{h}_1)\exp(s(\mathbf{m}_0, \mathbf{m}_1))}{\Gamma(\mathbf{h}_0, \mathbf{h}_1)\exp(s(\mathbf{m}_0, \mathbf{m}_1)) + \sum_{k \neq 0,1}(1 - \Gamma(\mathbf{h}_0, \mathbf{h}_k))\exp(s(\mathbf{m}_0, \mathbf{m}_k))}$$
$$+ \log \frac{\exp(s(\mathbf{p}_0, \mathbf{p}_1))/\Gamma(\mathbf{h}_0, \mathbf{h}_1)}{\exp(s(\mathbf{p}_0, \mathbf{p}_1))/\Gamma(\mathbf{h}_0, \mathbf{h}_1) + \sum_{k \neq 0,1}\exp(s(\mathbf{p}_0, \mathbf{p}_k))/(1 - \Gamma(\mathbf{h}_0, \mathbf{h}_k))},$$
(10)

where $\Gamma(\mathbf{h}_0, \mathbf{h}_1) = \exp(-d_\pi(\mathbf{h}_0, \mathbf{h}_1)/\beta)$ is the weight given by PEM. $\beta$ controls the sensitivity of similarity measure to PEM $d_\pi$. $s(\mathbf{u}, \mathbf{v}) := \frac{\mathbf{u}^T \mathbf{v}}{||\mathbf{u}||||\mathbf{v}||}$ denotes the cosine similarity function.

## 5 EXPERIMENTAL EVALUATION

We assess the effectiveness of our approach, Amortized Network Intervention (ANI), in managing networked temporal dynamics through simulated and real-world experiments. Our results demonstrate that ANI successfully reduces the mutual influence effects in both synthetic and two real datasets. We measure this improvement by calculating reduced intensities.

### 5.1 NETWORK INTERVENTION ON SYNTHETIC DATA

In our synthetic data experiments, we tested the proposed model on low-dimensional synthetic Multi-variate Hawkes Processes (MHP) without applying network amortization. To assess the performance

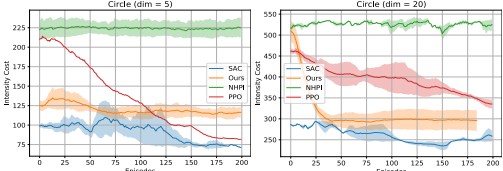 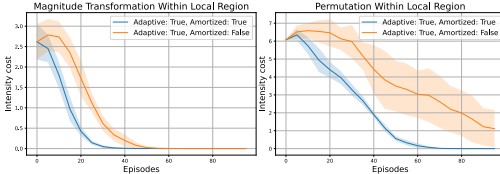

Figure 3: Cumulative intensity cost on synthetic datasets.

Figure 4: Generalization results of local community transformations on Covid data.

of our model-based reinforcement learning algorithm for dynamic network intervention, we compared against two model-free RL baselines, SAC (Haarnoja et al., 2018) and PPO (Schulman et al., 2017), as well as one model-based RL baseline called Neural Hawkes Process Intervention (NHPI) (Qu et al., 2023). We also adapted model-free RL techniques to TPP (Upadhyay et al., 2018) for event intervention and maintained the event intervention settings for NHPI to explore and compare the effectiveness of event intervention versus action intervention with high-frequency event data. Details on data generation for the synthetic dataset can be found in Appendix I.1.

Our study results are depicted in Fig. 3. Here, intensity cost means the average intensity throughout a fixed time period and over all the nodes within a local region. We later use this indicator to reflect the growth rate of the pandemic and traffic within a multivariate system. Remarkably, our approach achieves comparable levels of intensity reduction as SAC and PPO in both datasets, all without direct interaction with the environment. NHPI, which focuses on event intervention, faces difficulties in reducing activity intensity, especially with high-frequency event sequences. For additional generalization results on unseen MHPs with synthetic data, please refer to Appendix I.2.

## 5.2 EVALUATING GENERALIZATION ON COVID DATA

Our goal is to design an amortized city lock-down strategy that shares a similar policy structure for distinct city regimes to curb the epidemic by intervening in the influence matrix between cities. Concretely, we trained an amortized policy from five different county corpus and tested the amortized interventions on multiple unseen county dynamics. To generalize to an unseen split, the agent needs to be invariant to the orders of different counties and the amplitude or the phase of the spikes of the underlying excitatory point processes. Thus, we evaluated the generalization abil-

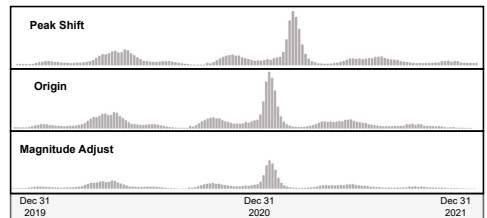

Figure 5: Two types of transformation of Covid data.

ity to the unseen counties corpus in two parts, local community transformation, and cross-community adaption, where local community transformation captures the agent's ability to generalize to a permuted or intensity-adjusted community, and cross-region adaption characterizes the ability to generalize to a intensity-peak-shifted community. We illustrate the two types of transformation in Fig. 5.

**Generalization Over Local Community Transformation** We show the generalization ability to a permutated or intensity-adjusted community by permutating and changing the intensity magnitude on the same community region and applying different control strategies to them. Fig. 4 demonstrates that amortized policy has a faster convergence rate than non-amortized policy on two types of transformed communities.

**Generalization Over Cross Community Adaption** We investigate how well the proposed approach generalizes over unseen intensity dynamics from different counties (w/ and w/o peak-shift). We evaluate the generalization performance in different county corpus with or without a similar dynamic structure to the training environment. Specifically, we define the testing environment as "in-distribution" or "generalize via interpolation" when the testing environment shares a similar intensity peak to the training environment and define the testing environment as "out-of-distribution" or "generalize via extrapolation" when the testing environment has a peak-shift or a delay effect to

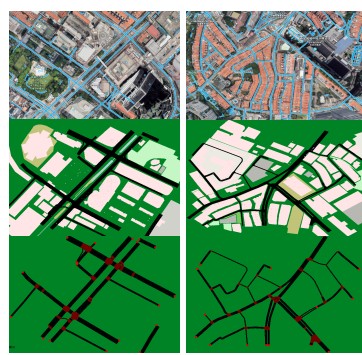

Figure 6: **Top**: Satellite map extracted from Google Earth (Goo, 2022). **Middle**: Road Network in SUMO (Lopez et al., 2018). **Bottom**: Extracted networks where red nodes are junction points.

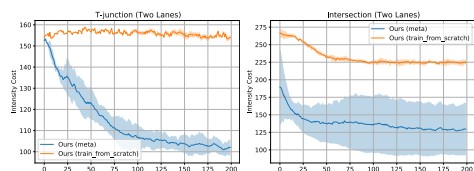

Figure 7: Generalization results for steering covid data on cross-community dynamics.

the original training environment. Table 1 summarizes the average reduced intensity for different methods under different region settings.

Notably, Table 1 ($1^{st}$ row) indicates that the non-adaptive and non-amortized policies are struggling to control the intensities in both in-distribution and out-of-distribution environments. Importantly, when we use an adaptive but non-amortized policy, the reduced intensities are quite obvious (Table 1, $3^{rd}$ row). This is not surprising since adaptively learning a policy (i.e., repeatedly updating the model with new policies) allows the agent to explore more possibilities in the environment and thus can obtain an optimal trajectory more easily. It is also interesting to point out that in-distribution environments are easier to generalize than out-of-distribution environments which contain a peak-shift or other complex transformations when compared with the trained environment. These findings are also consistent with the intensity cost curves illustrated in Fig. 7. We also provide an empirical study of two soft constraints on COVID data (i.e., limited intervention budget and limited intervention frequency) and additional baselines in Appendix J.2.

## 5.3 EVALUATING GENERALIZATION ON TRAFFIC DATA

We endeavored to enact network interventions aimed at alleviating traffic congestion within the urban road network system, particularly at road intersections. Event data were collected through SUMO (Lopez et al., 2018) simulations, whereby a traffic car was categorized as contributing to congestion if its velocity dropped below 0.5m/s. The network topology was derived from real-world cartography, as illustrated in Fig. 6, and subsequently processed by SUMO to create four distinct crossroad types (detailed information available in Appendix K.1). Following training on these crossroads, we assessed the generalization capabilities of our proposed amortized network intervention method on an additional set of four previously unseen road intersections. As depicted in Fig. 8, our results indicate that the learned meta-policy exhibits rapid adaptability to unfamiliar road systems only after a few gradient steps, demonstrating superior traffic congestion mitigation ability compared to a train-from-scratch model. Furthermore, we include a visual representation of the learned network intervention in Appendix K.3.

Figure 8: Generalization results of mitigated traffic flow on two unseen intersections from SUMO.

## 5.4 UNDERSTANDING GAINS FROM PEM: ABLATIONS AND VISUALIZATIONS

We show the efficacy of the proposed Policy Equivalent Embeddings (PEE) which are Bi-contrastive metric embeddings (Bi-CMEs) learned with Policy Equivalent Metrics (PEM) on the latent states by comparing it to Policy Similar Embeddings (PSE) (Agarwal et al., 2021) which is another common

Table 1: Reduced amount of intensities after network interventions for each node per unit time on four unseen communities on COVID data by different methods. We report average performance across 100 runs for three different seeds, with a standard deviation between parentheses.

| Adaptive | Amortized | Reduced Intensities | | | |
|---|---|---|---|---|---|
| | | In-distribution | | Out-of-distribution | |
| | | Georgia-0 | Alabama-0 | Georgia-1 | West Virginia-0 |
| False | False | -0.05(0.18) | 0.08(0.06) | -0.07(0.11) | -0.02(0.05) |
| | True | 0.21(0.43) | 0.18(0.58) | 0.06(0.24) | 0.02(0.02) |
| True | False | 0.18(0.19) | 0.14(0.22) | 0.02(0.13) | 0.15(0.10) |
| | True | **0.47(0.14)** | **0.71(0.42)** | **0.39(0.27)** | **0.54(0.27)** |

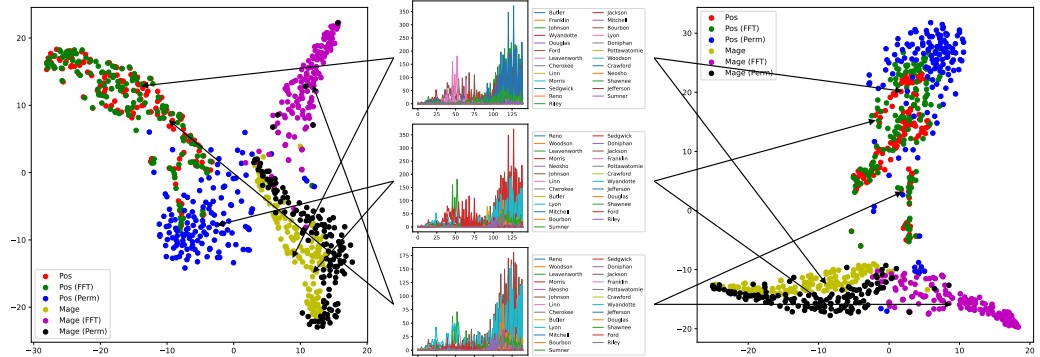

Figure 9: **Left:** t-SNE of latent embeddings learned with PEM. **Middle:** Original dynamic of selected 25 counties in Kansas (*Top*). Perturbed dynamics by two different transformations: random permutation and magnitude adjustment by FFT on the latent space $\mathbf{h}_t$ (*Middle, Bottom*). **Right:** t-SNE of latent embeddings learned with PSM. PEM successfully disentangle permutation-sensitive position embeddings (blue) and value-sensitive magnitude embeddings (purple).

generalization approach effective on pixel-based RL tasks. Specifically, we investigate the gains from Bi-CMEs and PEM by ablating them. Instead of learning Bi-CMEs jointly through the position and magnitude embeddings, we learn a separate CME for (Chen et al., 2020b) position and magnitude embeddings and use these separately learned embeddings to generate the policies.

Table 2 shows that PEEs (= PEM + Bi-CMEs) generalize significantly better than PSM or Single CMEs, both of which significantly degrade performance (-90%). This is not surprising since policy similar metric (PSM) requires two similar states collected by nearest neighbors which may introduce incorrect clusters on the latent state space. However, by introducing permutation equivalence as an inductive bias to the problem of controlling a dynamic system modeled by neural ODEs, PEM can better characterize the invariance features from different dynamic systems.

Table 2: **Ablation studies**. Reduced intensity after network interventions on West Virginia (Split 0) when we ablate the similarity metric and learning procedure for metric embeddings in different data augmentation settings. Each ablation entry is repeated for 100 trials for a fair comparison.

| Metric / Embedding | CMEs (Perm.) | CMEs (Magn.) | Bi-CMEs |
|---|---|---|---|
| PSM | 0.02(0.04) | 0.04(0.02) | 0.05(0.02) |
| PEM | 0.01(0.06) | 0.05(0.02) | **0.54(0.27)** |

**Visualizing learned representations** We visualize the metric embeddings in the ablation above by projecting them to two dimensions with t-SNE. Fig. 9 shows that PEEs partition the latent embeddings into four parts: (1) original position embeddings (red) and position embeddings with adjusted magnitude (green); (2) original magnitude embeddings (yellow) and magnitude embeddings with position permuted randomly (blue); (3) position embeddings with position permuted randomly (blue) which are orthogonal to the original position embeddings (red) and (4) magnitude embeddings with adjusted magnitude (purple) which are orthogonal to the original magnitude embeddings (yellow). Nevertheless, the projection of embeddings learned with PSM (Right in Fig. 9) gives a clear collapsing effect on position embeddings with position permuted randomly (blue) and magnitude embeddings with adjusted magnitude (purple). This finding is consistent with the results in Table 2 that Bi-CMEs weighted by PSM fail to extract permutation invariant and magnitude invariant information from the latent dynamic system.

## 6 CONCLUSIONS

This paper presents Amortized Networks Intervention, a versatile framework to steer the excitatory point processes. Our approach handles large-scale network interventions on a combinatorial action space, and achieves promising performance on challenging tasks on large, real-world datasets. Furthermore, the framework discussed here holds the potential for addressing significant problems like traffic light scheduling in urban areas.

ACKNOWLEDGMENTS

Shuang Li's research was in part supported by the NSFC under grant No. 62206236, Shenzhen Science and Technology Program under grant No. JCYJ20210324120011032, National Science and Technology Major Project under grant No. 2022ZD0116004, Shenzhen Key Lab of Cross-Modal Cognitive Computing under grant No. ZDSYS20230626091302006, and Guangdong Key Lab of Mathematical Foundations for Artificial Intelligence.

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

## A    PERMUTATION EQUIVALENT PROPERTY

**Definition 1** (Permutation Equivalent Policy). *Given a state $\mathbf{h}^t = (\mathbf{h}_1^t; \ldots; \mathbf{h}_N^t)$ and an action parameterized by a $k$-hot adjacency matrix in $\mathbb{R}^{N \times N}$, we say a policy is **permutation equivalent** (PE) to the state $\mathbf{h}^t$ if the order of corresponding rows in the adjacency matrix is also permuted accordingly when we reshuffle the orders the $N$ latent states. Mathematically, the permutation equivalent policy can be described by a function $\pi : \mathbb{R}^{N \times D} \to \mathbb{R}^{N \times N}$ such that*

$$\pi(\mathbf{h}^t) = \mathbf{P}^T \pi(\mathbf{P}\mathbf{h}^t)\mathbf{P}, \tag{11}$$

*where $\mathbf{P} \in \mathbb{R}^{N \times N}$ is any permutation matrix.*

**Definition 2** (Permutation Equivalent Metric, PEM). *For any $\mathbf{x}, \mathbf{y} \in \mathcal{S}$, where $\mathbf{y}$ is permuted state of $\mathbf{x}$, i.e., $\mathbf{y} = \mathbf{P}\mathbf{x}$, for some permutation matrix $\mathbf{P}$, the PEM under a distance $d$ and policy $\pi$ is described by $d_\pi : \mathcal{S} \times \mathcal{S} \to \mathbb{R}$, satisfying the recursive equation:*

$$d_\pi(\mathbf{x}, \mathbf{y}) = d(\pi(\mathbf{x}), \mathbf{P}^T \pi(\mathbf{y})\mathbf{P}) + \gamma d_\pi(\mathbf{x}', \mathbf{y}'), \tag{12}$$

*where $\mathbf{x}'$ and $\mathbf{y}'$ are the transition states of $\mathbf{x}$ and $\mathbf{y}$, given the deterministic dynamic $f$ and policy $\pi$.*

The proposed distance $d_\pi$ captures permutation equivalent behavior through a short-term distance $d$ between policy $\pi(\mathbf{x})$ and permuted policy $\pi(\mathbf{y})$, along with a long-term behavioral difference by recursive terms. The exact weights assigned to the two are given by the discount factor $\gamma$. Importantly, $d_\pi$ can be efficiently computed by approximate dynamic programming algorithms.

### A.1    DISTINCTION BETWEEN POLICY EQUIVALENCE AND POLICY INVARIANCE

We want to highlight the difference between presented policy equivalence and classic policy invariance as an inductive bias in RL literature. Previous research (Tang & Ha, 2021; Wang et al., 2020; Liu et al., 2020; Li et al., 2021), emphasize on a permutation invariant policy, i.e., $\pi(a|s) = \pi(a|\kappa(s))$ while our works introduce a permutation equivalent policy, i.e., $\pi(a|s) = \pi(\kappa(a)|\kappa(s))$, where $\kappa(\cdot)$ is any permutation mapping. Policy invariant property has wide applications on homogeneous agents to simplify collective behavior from complex cellular systems (Tang & Ha, 2021; Wang et al., 2020; Liu et al., 2020; Li et al., 2021). However, it cannot characterize the policy of an inhomogeneous system where the action should be permuted along with the states. In our case, the policy corresponds to an intervention matrix $A$ on a network $\mathcal{G}$ and thus a permuted network $\kappa(\mathcal{G})$ should also generate an '*equivalent*' permuted policy $\kappa(A)$. Thus, we name this property as '*policy equivalence*' to distinguish it from '*policy invariance*' from the literature.

## B    RELATED WORK

As previously highlighted in the introduction, several critical bottlenecks restrict universal applications of prompt control algorithms to disease. When it comes to quickly acquiring new skills, meta-learning emerges as an ideal paradigm for achieving rapid mastery in specific scenarios. In terms of data efficiency, both model-based reinforcement learning (MBRL) and meta-learning have the potential to significantly reduce sample complexity.

**Neural Temporal Point Process.** In the realm of modeling real-world data, the use of constrained models like Multivariate Hawkes Processes (Hawkes, 1971) can often lead to unsatisfactory results due to model misspecification. In recent studies, researchers have started exploring neural network parameterizations of Temporal Point Processes (TPPs) to mitigate these limitations. Common approaches (Du et al., 2016; Mei & Eisner, 2017) involve the employment of recurrent neural networks to evolve a latent state from which the intensity value can be derived. However, this approach falls short in capturing clustered and bursty event sequences, which are prevalent, as it overlooks vital temporal dependencies or necessitates an excessively high sampling rate (Nickel & Le, 2020).

To surmount these challenges, Neural Jump SDEs (Jia & Benson, 2019) and Neural Spatial Temporal Process (NSTT) (Chen et al., 2020a) extend the Neural Ordinary Differential Equation (ODE) framework, facilitating the computation of exact likelihoods for neural TPPs while addressing the limitations of prior methodologies. These two advancements closely align with our dynamic model, and we draw upon their concepts to develop neural excitatory point processes (EPPs) that are governed by an influence matrix. Besides Neural ODE, other streams of research on networked excitatory point processes are also proposed. Hawkes Processes on Large Network (Delattre et al., 2016) extends the construction of multivariate Hawkes processes to encompass a potentially infinite network of counting processes situated on a directed graph. Other approaches of latent structure

learning in multivariate point process (Cai et al., 2022; Fang et al., 2023) emphasize accommodating heterogeneous user-specific traits and incorporating both excitatory and inhibitory influences.

**Manipulation of Dynamic Processes.** The manipulation and control of dynamic processes represent an active area of research. Typically, control policies for temporal process manipulating can be divided into two categories: (1) Gradually introducing exogenous *event interventions* into the existing historical events, and (2) Promptly enforcing *network interventions* to the influence matrix between different types of events. Currently, most research is centered around the first type of intervention, primarily focusing on low-frequency and low-dimensional event interventions within social media datasets. For example, techniques like dynamic programming (Farajtabar et al., 2014; 2017) and stochastic control on SDE (Wang et al., 2018) with a closed feedback loop are utilized to steer user activities in social media platforms. However, the number of event types in the above work is limited to derive the closed-form solution. Modern Reinforcement Learning approaches, including both model-free (Upadhyay et al., 2018) and model-based RL (Qu et al., 2023), are proposed to mitigate fake news events on social media. Notably, the *event-intervention-based* approach will fail to generalize to high frequency and uncontrollable event data like newly infested disease cases and incoming traffic.

On the other hand, the problem of *network intervention*, particularly node manipulation (e.g., vaccination) to control epidemic processes on graphs has received extensive attention (Hoffmann et al., 2020; Medlock & Galvani, 2009). Most previous studies have adopted a static setup and made a single decision. In recent work (Meirom et al., 2021), the agent performs sequential decision-making to progressively control graph dynamics through node interventions, demonstrating effectiveness in slowing the spread of infections among different individuals. While existing *network-intervention-based* approaches offer a promising solution for individual-level quarantine in pandemic control, they do not inherently adapt to county- or state-level control, which necessitates more complex node status considerations than those assumed in (Meirom et al., 2021), as well as a larger search space incorporating edge interventions. Moreover, it's worth noting that our approach is related to, but more comprehensive than, epidemic control problems, as it accommodates various data distributions, including Poisson, within excitatory point processes.

**Model-based Reinforcement Learning.** The key to applications within a Reinforcement Learning (RL) framework lies in enhancing sample efficiency, with Model-Based Reinforcement Learning (MBRL) serving the role of approximating a target environment for the agent's interaction. In environments characterized by unknown dynamics, MBRL can either learn a deterministic mapping or a distribution of state transitions, denoted as $p(\Delta s|[s, a])$. Typically, modeling uncertainty in dynamic systems involves the evolution of a hidden unit to represent a dynamic world model. Several auto-regressive neural network structures are prevalent in this domain, including well-known models such as World Models Ha & Schmidhuber (2018), Decision Transformer (Chen et al., 2021), and the Dreamer family (Hafner et al., 2019; 2023). On the other hand, the integration of Neural Networks with Model Predictive Control (MPC) (Nagabandi et al., 2018) has achieved excellent sample complexity within model-based reinforcement learning algorithms. Our approach closely aligns with MPC and MBRL techniques, wherein MBRL methods, such as Gradient Descent, can be leveraged to refine or adapt the model utilized by MPC. This adaptation holds the potential to enhance performance, especially in scenarios where system dynamics are non-linear, partially unknown, or subject to change.

**Meta Reinforcement Learning.** The majority of meta Reinforcement Learning (RL) algorithms adhere to a model-free approach and introduce task-specific variational parameters to facilitate learning across various simple locomotion control tasks. Examples of such algorithms include MAESN (Gupta et al., 2018) and PEARL (Rakelly et al., 2019). Simultaneously, there has been notable success in recent times by incorporating the inherent sequential structure of off-policy control into the representation learning process, as demonstrated in works like CURL (Laskin et al., 2020), Sensory (Tang & Ha, 2021), and PSM (Agarwal et al., 2021) particularly when dealing with more complex tasks such as those found in the Distracting DM Control Suite (Stone et al., 2021).

In scenarios where data is limited, such as in disease control, researchers are increasingly focusing on Model-Based Meta Reinforcement Learning (MBMRL), with a specific emphasis on achieving *fast adaptation* within dynamics models. Approaches like AdMRL (Lin et al., 2020) and Amortized Meta Model-based Policy Search (AMBPS) (Wang & Van Hoof, 2022) involve the optimization and inference of task-specific policies within a parameterized family of tasks, often containing parameters related to positions and velocities. Importantly, a key distinction between our method and AMBPS

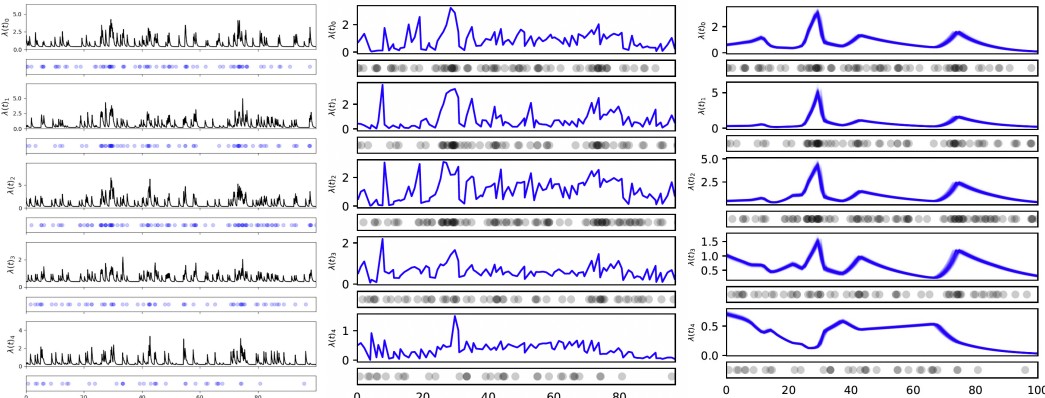

Figure 10: Temporal Point Process Model learned by different methods. **Left**: ground truth intensity function. **Middle**: Learned intensity function plot by Neural Jump ODE (Log-Likelihood: -33). **Right**: Learned intensity function plot by Neural Process. (Log-Likelihood: -183)

lies in our utilization of network embeddings and the incorporation of inductive bias to facilitate the learning of a meta-policy without parameterizing individual tasks.

## C   LIMITATIONS

It should be noted that our approach assumes that decisions made by considering a receding horizon window size of $T$ for each node provide a reasonably good approximation of the optimal policy. However, it is important to acknowledge that if long-range correlations exist, this approximation may result in decreased performance. Consequently, an intriguing question arises regarding the ability of our approach to effectively tackle the problem of network interventions within long-range correlated data distributions under the proposed training protocol.

## D   A COMPARISON BETWEEN NEURAL ODE AND NEURAL PROCESS

In this section, we conduct a comparative analysis of model performance and efficiency between Neural Ordinary Differential Equations (Neural ODE or NODE) (Chen et al., 2018) and Neural Process (NP) (Garnelo et al., 2018). To evaluate these models, we utilized a synthetic dataset consisting of 5-dimensional Multi-Hyperparameter (MHP) data, with each dimension containing 100 sample points in the training set. The results of this comparison are visually presented in Fig. 10.

Our observations reveal notable distinctions between the learned behaviors of NODE and the Neural Process model. Specifically, the curve learned by NODE exhibits greater diversity, as depicted in the middle panel of Fig. 10. An important metric to consider is the log-likelihood, where we find that Neural ODE achieves a log-likelihood of -33, significantly outperforming Neural Process with a log-likelihood of -180.

It's essential to consider the computational cost associated with each model. For Neural ODE, a significant portion of the computational burden arises from the numerical integration process itself. The runtime complexity of this integration process is denoted as $O(\text{NFE})$, where NFE represents the number of function evaluations. The worst-case scenario for NFE depends on two factors: the minimum step size $\delta$ required by the ODE solver and the maximum integration time of interest, denoted as $\Delta t_{max}$.

In contrast, the runtime complexity of Neural Process, which employs $n$ context points and $m$ target points, is represented as $O(n + m)$. In our specific experiment, the number of total function evaluations (NFE) for NODE was approximately 5000, while for Neural Process, we selected 50 context points and 100 target points.

It is worth noting that the integration steps, as quantified by $\frac{\Delta t_{max}}{\delta}$, can potentially result in a large constant factor, which may be hidden within the big-O notation. However, it is reassuring to acknowledge that modern ODE solvers, such as *dorpi5*, are designed with adaptive step size mechanisms that adjust dynamically to the supplied data. This adaptive behavior mitigates concerns related to the scalability of the integration process with respect to dataset size and complexity.

# E  TECHNICAL DETAILS OF A PROBABILISTIC NETWORKED MODEL

Table 3: Table of conditional spike count distributions, their parameterizations, and their properties

| Distribution | $p(x\|\psi, \upsilon)$ | $\mathbb{E}(x)$ | $\text{Var}(x)$ |
|---|---|---|---|
| $\text{Poi}(exp(\psi))$ | $\frac{exp(-exp(\psi))(exp(\psi))^x}{x!}$ | $exp(\psi)$ | $exp(\psi)$ |
| $\text{Bern}(\sigma(\psi))$ | $\sigma(\psi)^x \sigma(-\psi)^{1-x}$ | $\sigma(\psi)$ | $\sigma(\psi)\sigma(-\psi)$ |
| $\text{Bin}(\upsilon, \sigma(\psi))$ | $\binom{\upsilon}{x}\sigma(\psi)^x\sigma(-\psi)^{\upsilon-x}$ | $\upsilon\sigma(\psi)$ | $\upsilon\sigma(\psi)\sigma(-\psi)$ |
| $\text{NB}(\upsilon, \sigma(\psi))$ | $\binom{\upsilon+x-1}{x}\sigma(\psi)^x\sigma(-\psi)^\upsilon$ | $\upsilon \cdot exp(\psi)$ | $\upsilon exp(\psi)/\sigma(-\psi)$ |

# F  TECHNICAL DETAILS OF MEAN FIELD APPROXIMATION FOR REWARD MODEL

We begin by defining some notation which will be used throughout these results

- Spectrum of a nonlinear operator $f$: $L_f = ||f||_{Lip} := \sup_{x \neq y} \frac{||f(x)-f(y)||}{||x-y||} < \infty$

## F.1  THEORETICAL JUSTIFICATION

We use $\mathbf{x}^i = \{x_1^i, \ldots, x_N^i\}$ to record the incremental number of events for different nodes starting from within $[\tau_{i-1}, \tau_i)$. We consider a stochastic dynamic system based on the nonlinear system Eq. (2-4) presented in the manuscript:

$$\mathbf{h}_n^{\tau_0} = \mathbf{h}_n^0$$
$$\frac{d\mathbf{h}_n^\tau}{d\tau} = f(\tau, \mathbf{h}_n^\tau), \ \ \forall \tau \in \mathbb{R}_+ \setminus \cup_i \{\tau_i\},$$
$$\lim_{\epsilon \downarrow 0} \mathbf{h}_n^{\tau_i + \epsilon} = \sum_{m \in \mathcal{N}_n} w_{m \to n} \cdot \phi(\mathbf{h}_m^{\tau_i}, \mathbf{x}_m^i). \tag{13}$$

where $f : \mathbb{R}^d \to \mathbb{R}^d$ and $\phi : \mathbb{R}^d \to \mathbb{R}$ are two Lipschitz functions.

The emission/intensity function for the dynamic is defined as $\Lambda(i|\mathbf{h}^{\tau_i-}) := g_\Lambda(\mathbf{h}^{\tau_i-} \cdot \mathbf{w}) = g_{\Lambda, \mathbf{w}}(\mathbf{h}^{\tau_i-})$ where $g_{\Lambda, \mathbf{w}} : \mathbb{R}^{N \times d} \to \mathbb{R}^N$ is a Lipschitz activation function, $\mathbf{w} \in \mathbb{R}^d$ is a linear layer, and $\mathbf{h}^{\tau_i-} \in \mathbb{R}^{N \times d}$ is the left continuous points before jump. We use $\lambda_m^i$ to denote $m$-th element of $\Lambda(i)$ and thus $\mathbf{x}_m^i \sim \text{Poi}(\lambda_m(i|\mathbf{h}_m^{\tau_i-}))$ is the number of incremental events in $[\tau_{i-1}, \tau_i)$ for node $m$. The ground truth cumulative cost $J(t)$ at finite horizon time $t$ is given by

$$J(t) := \sum_{i=0}^t \gamma^i (\sum_{n=1}^N \mathbb{E}[x_n^i]). \tag{14}$$

Instead of extensively performing Monte Carlo simulation to obtain the cost estimator, we introduce a ***Mean Field Estimator*** $\hat{J}(t)$ for $J(t)$ by averaging out the interaction effect by $x_m^i$ in the dynamic system (13), i.e.,

$$\hat{J}(t) := \sum_{i=0}^t \gamma^i (\sum_{n=1}^N \hat{\lambda}_n^i). \tag{15}$$

where $\hat{\lambda}_n^i := \lambda_n(i|\hat{\mathbf{h}}_n^{\tau_i-})$ and $\hat{\mathbf{h}}_n^{\tau_i-}$ is the mean-field estimator for $\mathbf{h}_n^{\tau_i-}$ and is derived from a deterministic dynamic by replacing the stochastic discrete update term in system (13) with $\phi(\mathbf{h}_m^{\tau_i}, \hat{\lambda}_m^i)$. We let the initial states be the same point, i.e., $\hat{\mathbf{h}}_n^0 = \mathbf{h}_n^0, \forall n \in \{1, 2, \cdots, N\}$.

**Proposition 1.** *Given $f$ and $g$ are Lipschitz in Sys. (13), let $\mathbf{h}_n^{\tau_i^-}$ be the left continuous point of $\mathbf{h}_n^{\tau_i}$ in the Sys. (13), then it can be recursively expressed by a **Lipschitz** operator $\mathcal{T}_n : \mathbb{R}^{N \times d} \times \mathbb{R}^N \to \mathbb{R}^d$ :*

$$\mathbf{h}_n^{\tau_i^-} = \mathcal{T}_n(\mathbf{h}^{\tau_{i-1}^-}, \mathbf{x}^{i-1}) \tag{16}$$

*where $\mathbf{h}^{\tau_{i-1}^-} \in \mathbb{R}^{N \times d}$, and $\mathbf{x}^{i-1} \in \mathbb{R}^N$. More importantly, let $\lambda_w$ be the maximum spectrum of influence matrix $\mathbf{W}$, when $L_f$, $L_\phi$, and $\lambda_w$ are smaller than 1, we have $L_{\mathcal{T}_n} < 1$.*

*Proof.* We define $\Phi(\mathbf{h}^{\tau_i}, \mathbf{x}^i) := [\phi(\mathbf{h}_1^{\tau_i}, \mathbf{x}_1^i), \phi(\mathbf{h}_2^{\tau_i}, \mathbf{x}_2^i), \cdots, \phi(\mathbf{h}_N^{\tau_i}, \mathbf{x}_N^i)]^T \in \mathbb{R}^{N \times d}$ as the discrete kernel in System (13), we can represent $\mathbf{h}_n^{t_i^-}$ as:

$$\mathbf{h}_n^{\tau_i^-} = \mathcal{T}_n(\mathbf{h}^{\tau_{i-1}}, \mathbf{x}^{i-1}) = \mathbf{w}^T \Phi(\mathbf{h}^{\tau_{i-1}}, \mathbf{x}^{i-1}) + \int_{\tau_{i-1}^+}^{\tau_i^-} f(\mathbf{h}_n^s)\, ds,$$

where $\mathbf{w}^T$ is the $n$-th row of the influence matrix $\mathbf{W}$. Since $f$ and $\phi$ are Lipschitz and the composition of Lipschitz functions is also Lipschitz, so $\mathcal{T}_n$ is Lipschitz. Since $L_{\mathcal{T}} \leq \lambda_w \cdot (L_f)^m \cdot (L_\phi)^N$ where $m$ is the number of summation in the intergral term, we have $L_{\mathcal{T}_n} < 1$ when $L_f$, $L_\phi$, and $\lambda_w$ are smaller than 1.

$\square$

**Proposition 2.** *Given $f$ and $g$ are Lipschitz in Sys. (13), let $\hat{\mathbf{h}}_n^{\tau_i^-}$ be the left continuous point of $\hat{\mathbf{h}}_n^{\tau_i}$ in the deterministic version of the presented Sys. (13) by replacing $\mathbf{x}_m^i$ with $\hat{\lambda}_m^i$, then it can be recursively expressed by a* **Lipschitz** *operator $\mathcal{T}_n : \mathbb{R}^{N \times d} \times \mathbb{R}^N \to \mathbb{R}^d$ :*

$$\hat{\mathbf{h}}_n^{\tau_i^-} = \mathcal{T}_n(\mathbf{h}^{\tau_{i-1}}, \hat{\Lambda}(i-1)) \tag{17}$$

*where $\hat{\Lambda}(i-1) = [\hat{\lambda}_1^{i-1}, \hat{\lambda}_2^{i-1}, \cdots, \hat{\lambda}_N^{i-1}]$.*

*Proof.* This is a direct result from Prop. 1. $\square$

**Lemma 1.** *Let $\hat{\mathbf{h}}_n^{\tau_i}$ be the mean field estimator for $\mathbf{h}_n^{\tau_i}$, suppose $f$ and $\phi$ are Lipshitz , and $\forall n, i$, $\lambda_n^i$ is bounded by $L_0$, we have:*

$$\mathbb{E}\Big[||\mathbf{h}_n^{\tau_i^-} - \hat{\mathbf{h}}_n^{\tau_i^-}||\Big] \leq L_{\mathcal{T}_n} \mathbb{E}\Big[||\mathbf{h}_n^{\tau_{i-1}^-} - \hat{\mathbf{h}}_n^{\tau_{i-1}^-}||\Big] + M_n, \tag{18}$$

*where $M_n = L_{\mathcal{T}_n}(L_0^{1/2} + L_0)$. Moreover, when $L_f < 1$ and $L_\phi < 1$, we have:*

$$\mathbb{E}\Big[||\mathbf{h}_n^{\tau_i^-} - \hat{\mathbf{h}}_n^{\tau_i^-}||\Big] \leq \Big(1 - (L_{\mathcal{T}_n})^i\Big) \cdot \frac{M_n}{1 - L_{\mathcal{T}_n}}. \tag{19}$$

*Proof.* For simplicity, we use $\mathbf{h}_n^i$ to denote $\mathbf{h}_n^{\tau_i^-}$ and $\hat{\mathbf{h}}_n^i$ to denote $\hat{\mathbf{h}}_n^{\tau_i^-}$. Here we prove the first inequality: By Prop. 1 and Prop. 2, we can decompose $\mathbf{h}_n^i$ and $\hat{\mathbf{h}}_n^i$, i.e.,

$$\mathbb{E}\Big[||\mathbf{h}_n^i - \hat{\mathbf{h}}_n^i||\Big] = \mathbb{E}\Big[||\mathcal{T}_n(\mathbf{h}^{i-1}, \mathbf{x}^{i-1}) - \mathcal{T}_n(\hat{\mathbf{h}}^{i-1}, \hat{\Lambda}(i-1))||\Big]$$

$$\leq \mathbb{E}\Big[L_{\mathcal{T}_n}||(\mathbf{h}^{i-1}, \mathbf{x}^{i-1}) - (\hat{\mathbf{h}}^{i-1}, \hat{\Lambda}(i-1))||\Big]$$

$$\leq \mathbb{E}\Big[L_{\mathcal{T}_n}\big[||\mathbf{h}^{i-1} - \hat{\mathbf{h}}^{i-1}|| + ||\mathbf{x}^{i-1} - \hat{\Lambda}(i-1)||\big]\Big]$$

$$= \mathbb{E}\Big[L_{\mathcal{T}_n}\big[||\mathbf{h}^{i-1} - \hat{\mathbf{h}}^{i-1}|| + ||\mathbf{x}^{i-1} - \Lambda^{i-1} + \Lambda^{i-1} - \hat{\Lambda}(i-1)||\big]\Big]$$

$$\leq L_{\mathcal{T}_n}\mathbb{E}\Big[||\mathbf{h}^{i-1} - \hat{\mathbf{h}}^{i-1}||\Big] + \mathbb{E}\Big[L_{\mathcal{T}_n}||\mathbf{x}^{i-1} - \Lambda(i-1)||\Big] + L_{\mathcal{T}_n}||\Lambda(i-1) - \hat{\Lambda}(i-1)||$$

$$= L_{\mathcal{T}_n}\mathbb{E}\Big[||\mathbf{h}^{i-1} - \hat{\mathbf{h}}^{i-1}||\Big] + L_{\mathcal{T}_n}\mathbb{E}\Big[||\mathbf{x}^{i-1} - \Lambda(i-1)||\Big] + L_{\mathcal{T}_n}||\Lambda(i-1) - \hat{\Lambda}(i-1)||$$

$$\leq L_{\mathcal{T}_n}\mathbb{E}\Big[||\mathbf{h}^{i-1} - \hat{\mathbf{h}}^{i-1}||\Big] + L_{\mathcal{T}_n} \cdot (L_0^{1/2} + L_0)$$

$$= L_{\mathcal{T}_n}\mathbb{E}\Big[||\mathbf{h}^{i-1} - \hat{\mathbf{h}}^{i-1}||\Big] + M_n$$

Given the above result, we can derive its fixed point $x^*$ i.e.,

$$x^* = L_{\mathcal{T}_n}x^* + M_n \tag{20}$$

$$x^* = \frac{M_n}{1 - L_{\mathcal{T}_n}} \tag{21}$$

Since $L_{\mathcal{T}_n} < 1$ given $L_f < 1$ and $L_g < 1$, Eq. (18) is a contraction and thus starting from any arbitrary point $x_k$ will converge to this fixed point $x^*$. So we have:

$$\mathbb{E}\Big[||\mathbf{h}_n^i - \hat{\mathbf{h}}_n^i||\Big] - \frac{M_n}{1 - L_{\mathcal{T}_n}} \leq L_{\mathcal{T}_n}\left( \mathbb{E}\Big[||\mathbf{h}_n^{i-1} - \hat{\mathbf{h}}_n^{i-1}||\Big] - \frac{M_n}{1 - L_{\mathcal{T}_n}} \right) \tag{22}$$

$$\leq (L_{\mathcal{T}_n})^i\left( \mathbb{E}\Big[||\mathbf{h}_n^0 - \hat{\mathbf{h}}_n^0||\Big] - \frac{M_n}{1 - L_{\mathcal{T}_n}} \right), \tag{23}$$

$$\mathbb{E}\Big[||\mathbf{h}_n^i - \hat{\mathbf{h}}_n^i||\Big] \leq \left( 1 - (L_{\mathcal{T}_n})^i \right) \cdot \frac{M_n}{1 - L_{\mathcal{T}_n}}, \tag{24}$$

where the last inequality follows by $\mathbf{h}_n^0 = \hat{\mathbf{h}}_n^0$. Thus $\mathbb{E}\Big[||\mathbf{h}_n^i - \hat{\mathbf{h}}_n^i||\Big]$ is bounded. $\qquad\square$

Next, let's prove **Theorem 1**.

*Proof.*

$$|J(t) - \hat{J}(t)| = \left| \sum_{n=1}^{N}\sum_{i=0}^{t} \mathbb{E}[x_n^i] - \sum_{n=1}^{N}\sum_{i=0}^{t} \hat{\lambda}_n^i \right| \tag{25}$$

$$= \left| \sum_{n=1}^{N}\sum_{i=0}^{t} \mathbb{E}_{\mathbf{h}_n^i}\mathbb{E}_{x_n^i}[x_n^i|\mathbf{h}_n^i] - \sum_{n=1}^{N}\sum_{i=0}^{t} g_{\lambda_n,\mathbf{w}}(\hat{\mathbf{h}}_n^i) \right| \tag{26}$$

$$= \left| \sum_{n=1}^{N}\sum_{i=0}^{t} \mathbb{E}_{\mathbf{h}_n^i} g_{\lambda_n,\mathbf{w}}(\mathbf{h}_n^i) - \sum_{n=1}^{N}\sum_{i=0}^{t} g_{\lambda_n,\mathbf{w}}(\hat{\mathbf{h}}_n^i) \right| \tag{27}$$

$$= \left| \sum_{n=1}^{N}\sum_{i=0}^{t} \mathbb{E}_{\mathbf{h}_n^i}\Big[ g_{\lambda_n,\mathbf{w}}(\mathbf{h}_n^i) - g_{\lambda_n,\mathbf{w}}(\hat{\mathbf{h}}_n^i) \Big] \right| \tag{28}$$

$$\leq \sum_{n=1}^{N}\sum_{i=0}^{t} \mathbb{E}_{\mathbf{h}_n^i}\Big[ |g_{\lambda_n,\mathbf{w}}(\mathbf{h}_n^i) - g_{\lambda_n,\mathbf{w}}(\hat{\mathbf{h}}_n^i)| \Big] \tag{29}$$

$$\leq \sum_{n=1}^{N}\sum_{i=0}^{t} \mathbb{E}_{\mathbf{h}_n^i}\Big[ L_{g_{\lambda_n},\mathbf{w}}||\mathbf{h}_n^i - \hat{\mathbf{h}}_n^i|| \Big] \tag{30}$$

Apply Lemma 1, we have,

$$\leq \sum_{n=1}^{N}\sum_{i=0}^{t} L_{g_{\lambda_n},\mathbf{w}} \cdot \left( 1 - (L_{\mathcal{T}_n})^i \right) \cdot \frac{M_n}{1 - L_{\mathcal{T}_n}} \tag{31}$$

$$\leq \sum_{n=1}^{N} (t+1) \cdot L_{g_{\lambda_n},\mathbf{w}} \cdot \frac{M_n}{1 - L_{\mathcal{T}_n}} \tag{32}$$

$$\leq N(t+1) \cdot L_{g_{\lambda_n},\mathbf{w}} \cdot \frac{\max_n M_n}{1 - \max_n(L_{\mathcal{T}_n})} \tag{33}$$

$$\square$$

### F.2 EMPIRICAL EVALUATION

We consider the setting of a simplified linear dynamic model based on the nonlinear system (2-4) presented in the manuscript. Specifically, the nonlinear ODE drift term $f(\mathbf{h}_n^\tau)$ is replaced by a linear function $\mathbf{A}\mathbf{h}_n^\tau + \mathbf{b}$ where $\mathbf{A} \in \mathbb{R}^{d\times d}$ and $\mathbf{b} \in \mathbb{R}^d$. The nonlinear discrete jump kernel $\phi(\mathbf{h}_m^{\tau_i}, x_m^i)$ is replaced by a linear kernel $C\mathbf{h}_m^{\tau_i} + Dx_m^i$, where $C \in \mathbb{R}$ represents history forgetness and $D \in \mathbb{R}$ is the scaling factor for the new events. Mathematically, the simplified linear dynamic is given in System (34).

$$\begin{cases} \mathbf{h}_n^{\tau_0} = \mathbf{h}_n^0, \\ \frac{d\mathbf{h}_n^\tau}{d\tau} = \mathbf{A}\mathbf{h}_n^\tau + \mathbf{b}, \\ \lim_{\epsilon\to 0} \mathbf{h}^{\tau_i+\epsilon} = \sum_{m\in\mathcal{N}_n} w_{m\to n}(C\mathbf{h}_m^{\tau_i} + Dx_m^i). \end{cases} \tag{34}$$

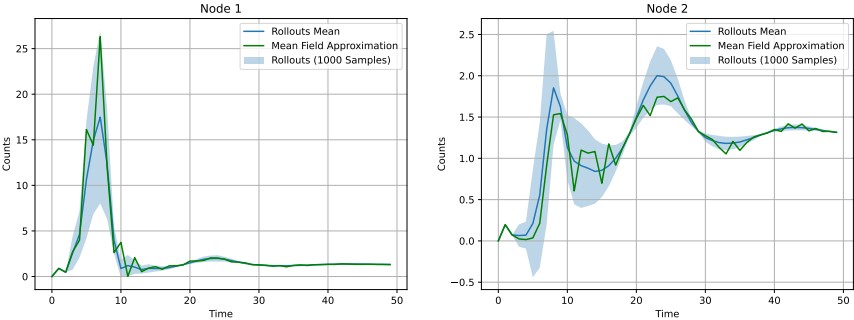

Figure 11: Empirical Evaluation of Mean Field Approximation

Empirically, we performed a toy experiment on the simplified 2-D linear dynamic and we found the mean field approximator provides an efficiency and accurate approximation for the rollout means (Fig. 11).

## G   DIFFERENCE BETWEEN SIMCLR AND THE PROPOSED METHOD

Given a random sampled minibatch of $N$ examples and the SimCLR contrastive prediction task is defined on pairs of augmented examples derived from the minibatch, resulting in $2N$ data points. SimCLR treats the other $2(N-1)$ augmented examples within a minibatch as negative examples. Then the loss function for a positive pair of examples $(i, j)$ is defined as,

$$l_{i,j} = -\log \frac{\exp(\text{sim}(\mathbf{z}_i, \mathbf{z}_j)/\tau)}{\sum_{k=1}^{2N} 1_{[k \neq i]} \exp(\text{sim}(\mathbf{z}_i, \mathbf{z}_k)/\tau)}, \tag{35}$$

where $\mathbf{z}_i$ is the projected embedding from augmented samples.

Compared with SimCLR, our proposed loss does not require sampling additional batch data as negative examples and projecting the augmented sample to one embedding, instead, we only use one anchor sample (anchor network) and project the sample to two different embeddings ($p$ and $m$), and augment the anchor sample by permuting its node orders or adding noise to the node embeddings such that it can form two negative pairs naturally (as depicted in Fig. 1). Moreover, we augment our network on its latent embedding space by utilizing the graph augmentation techniques, while SimCLR directly performs augmentations on the original visual representations.

## H   ARCHITECTURE AND HYPERPARAMTERS

For the dynamic model, we parameterized the ODE forward function $f_h$ in Eq. (3) as a Time-dependent multilayer perception (MLP) with dimensions [64-64-64]. We used the Softplus activation function. We first attempted to use MLP to parameterize the instantaneous jump function $\phi_h$ in Eq. (4); however, it led to unstable results for long sequences. Thus we switched to the GRU parameterization, which takes an input, the latent state $\mathbf{h}^{t_i}$, and outputs a new latent state. Importantly, we found letting $f_h$ and $\phi_h$ share all the parameters across different node trajectories will also lead to the failure of capturing the mode diversity. We therefore created an independent biased term added between the MLP layers in $f_h$ to compensate for diversity loss. Lastly, we use MLP to parameterize the emission function $g_\lambda$.

We initialized all Neural ODEs (for the hidden state) with zero drift by initializing the weights and biases of the final layer to zero. All integrals were solved using (Chen et al., 2018) to within a relative and absolute tolerance of 1E-4 or 1E-6, chosen based on preliminary testing for convergence and stability. We also use Seminorms (Kidger et al., 2021) to accelerate neural ODE learning and apply temporal regularization (Ghosh et al., 2020) to mitigate the effect of stiff ODE systems.

For the policy model, we parameterized the policy network by 4 heads, and 128 d-model Transformer layers. For the synthetic dataset, we used a 2-layer transformer for representation learning and another 2-layer transformer for policy generation. For the real-world dataset, we used a 4-layer transformer for representation learning and another 4-layer transformer for policy generation.

For the Policy Equivalent Metric presented in Definition 2, we learned a value function parameterized by a 2-layer transformer by optimizing Least Square Temporal Difference (LSTD).

We trained the dynamic, policy, and PEM-value network by using the ADAM optimizer with a 1E-2 decay rate across 4 RTX3090 GPUs. The initial learning rates for dynamic learning, policy learning, and PEM-value function learning were set to be 1E-3, 1E-4, and 1E-4 respectively.

# I   ADDITIONAL DETAILS OF SYNTHETIC DATA EXPERIMENT

## I.1   DATASET SETUP

We generated synthetic networked point process data by simulating Multivariate Hawkes processes (MHP), which are doubly stochastic point processes with self-excitations (Hawkes, 1971). Specifically, the underlying ground truth influence matrix $\mathbf{W}$ was generated with $n = 10$ nodes and the weights were set as $w_{ij} \sim \mathcal{U}[0, 0.5]$. We set the graph sparsity to 0.1, *i.e.*, each edge is kept with probability 0.1. The generated influence matrix was adjusted appropriately so that its maximum spectral radius was smaller than one, ensuring the stability of the process. We further set the Hawkes kernel to be an exponential basis kernel, where the parameter was set to $\beta = 4$, meaning roughly losing 98% of influence after one unit of time. The simulation of MHP was based on a thinning algorithm (Ogata, 1981) on a $T = 100$ horizon.

## I.2   ADDITIONAL RESULTS AND FIGURES

We additionally trained the amortized policy on a synthetic star graph and a circular graph with different ground-truth weight matrices and tested the performance on a new star or cycle graph with random weights. The results are shown in Fig. 12. Surprisingly, we find the amortized policy only slightly outperforms the non-amortized on both environments. We conjecture this is because we are adapting the amortized policy to a small local region (only 10 nodes) in this experiment so that the non-amortized policy already can achieve relatively good results.

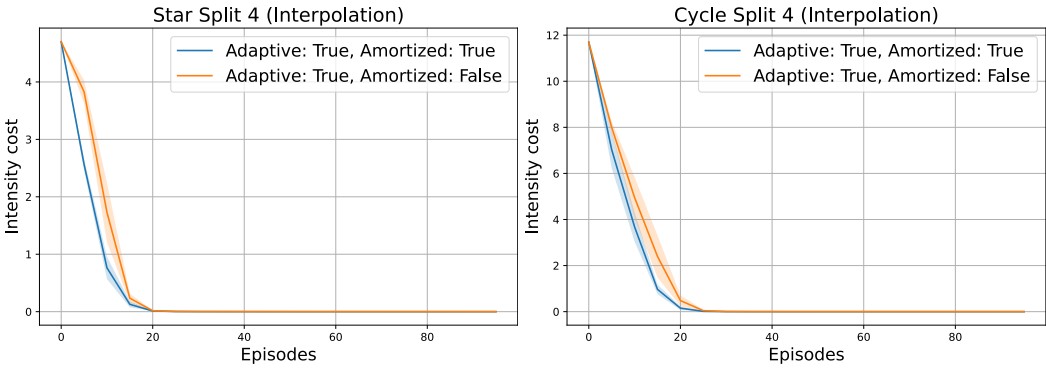

Figure 12: Generalization Results over synthetic data

# J   ADDITIONAL DETAILS OF COVID DATA EXPERIMENT

## J.1   DATASET SETUP

We used data released publicly by (NYTimes, 2020) on daily COVID-19 to learn the excitatory point processes of the pandemic outbreak. The data contains the cumulative counts of coronavirus cases in the United States, at the state and county level, over time. Specifically, we separated the U.S. COVID-19 data into state-wise records and further split a state-wise record into different county corpus where each split is named as "a local region" or "a split", containing distinct intensity trajectories from no more than 25 counties.

## J.2   ADDITIONAL RESULTS AND FIGURES

**Additional Baselines for COVID data** We applied two additional baselines SAC and PPO on a randomly chosen community that contains nine counties. We used the learned NJODE model as the covid environment simulator and the learned model (Without control) is shown in Fig. 13 along with the observed ground truth counts. We also attempted to use the plain Hawkes model (only influence matrix $A$ and baseline $b$ are learnable) to learn the underlying COVID dynamic. As Fig. 13 depicted, however, plain multivariate Hawkes processes (Purple) fail to distinguish the intensity difference between different counties while the NJODE model (Black) correctly captures the characteristic

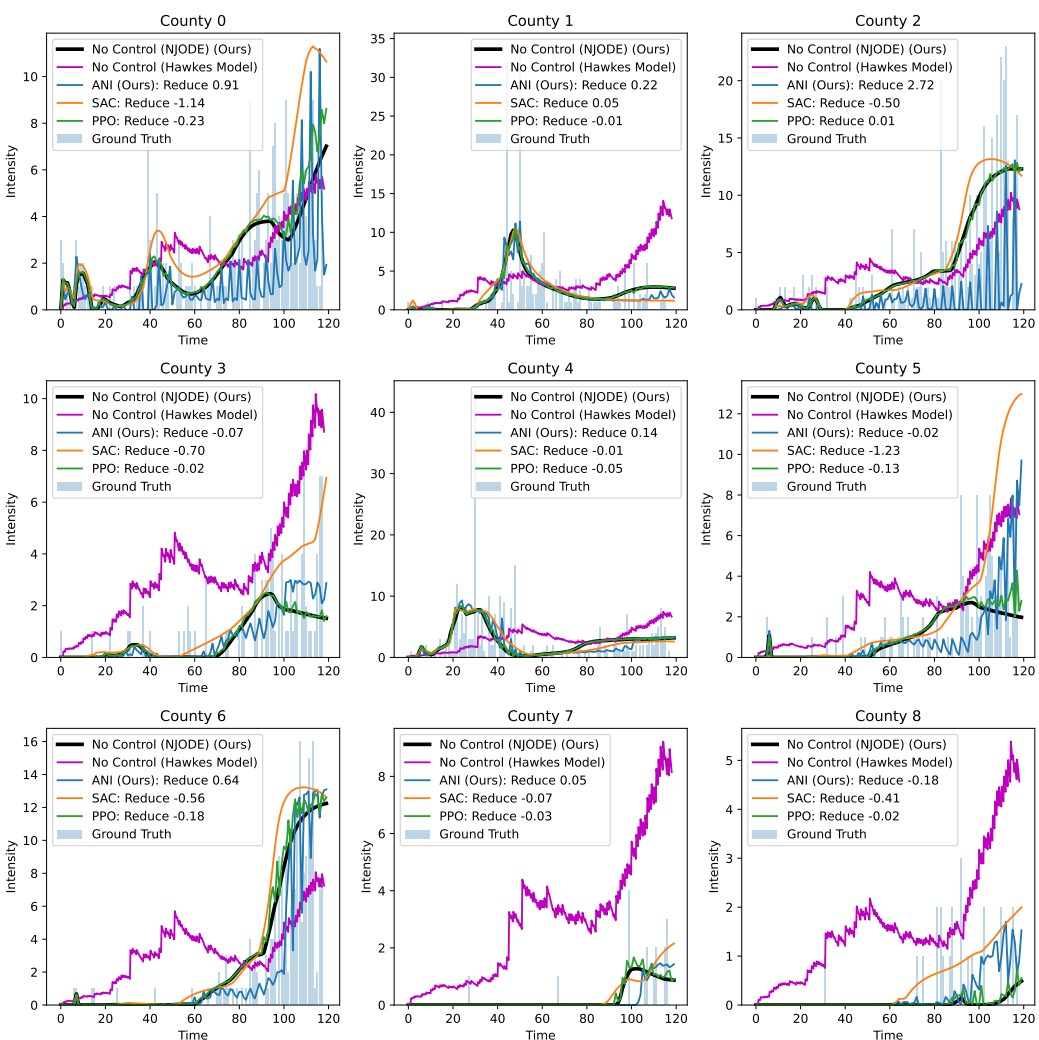

Figure 13: Additional baselines on nine counties

of different counties and also preserves multimodality within each county. We conjecture this is because the plain multivariate Hawkes process does not have enough parameters to characterize the individual variation on the complex COVID data dynamic correctly. Regarding the intervention effect, we observe that ANI has successfully reduced the number of infested people (having a positive reduced intensity value) in six counties among nine while SAC and PPO struggle to have a positive intervention effect on the nine counties. We speculate it is caused by the disconnected gradients between the consecutive latent states (without backpropagating through the learned COVID dynamic model itself) when using model-free RL algorithms like SAC and PPO.

**Fairness constraints and more** We also present the results and trade-offs for different policies under various fairness constraints. Specifically, we use $\lambda_1$ to control the weight of an intervention budget cost and use $\lambda_2$ to control the weight of a policy smoothing cost. The intervention cost used here is defined by the distance between two counties when an intervention is implemented and the smoothing cost is defined by the distance between two consecutive policies. We use the average reduced intensity and the average lockdown probability for each edge during the total horizon to measure fairness and the result is shown in Table 4. Interestingly, we find imposing heavy constraints on the policy simultaneously (*i.e.*, $\lambda_1 = 0.1, \lambda_2 = 0.1$) does not lead to the lowest lockdown probability (1.29%) but does give the lowest control effect (0.01). Instead, only enforcing smoothing constraints (*i.e.*, $\lambda_1 = 0.0, \lambda_2 = 0.1$) gives us a fairer policy, i.e., less effort to intervene (average lockdown probability: 0.43 %) but achieve fair control effect (average reduced intensity: 0.10). We conjecture that adding extra intervention cost constraints will discourage the agent from exploring

and thus underperform policy smoothing constraints. We illustrate the detailed lockdown for different constraints in Fig. 14.

Table 4: Average lockdown probabilities and reduced intensity under different soft constraints.

| Fairness / Parameters | $\lambda_1 = 0.0$ | | $\lambda_1 = 0.1$ | |
|---|---|---|---|---|
| | $\lambda_2 = 0.0$ | $\lambda_2 = 0.1$ | $\lambda_2 = 0.0$ | $\lambda_2 = 0.1$ |
| Average Reduced intensity | 0.25(0.06) | 0.10(0.04) | 0.06(0.05) | 0.01(0.03) |
| Average Lockdown probability (%) | 1.31(0.53) | 0.43(0.12) | 1.29(0.27) | 1.29(0.17) |

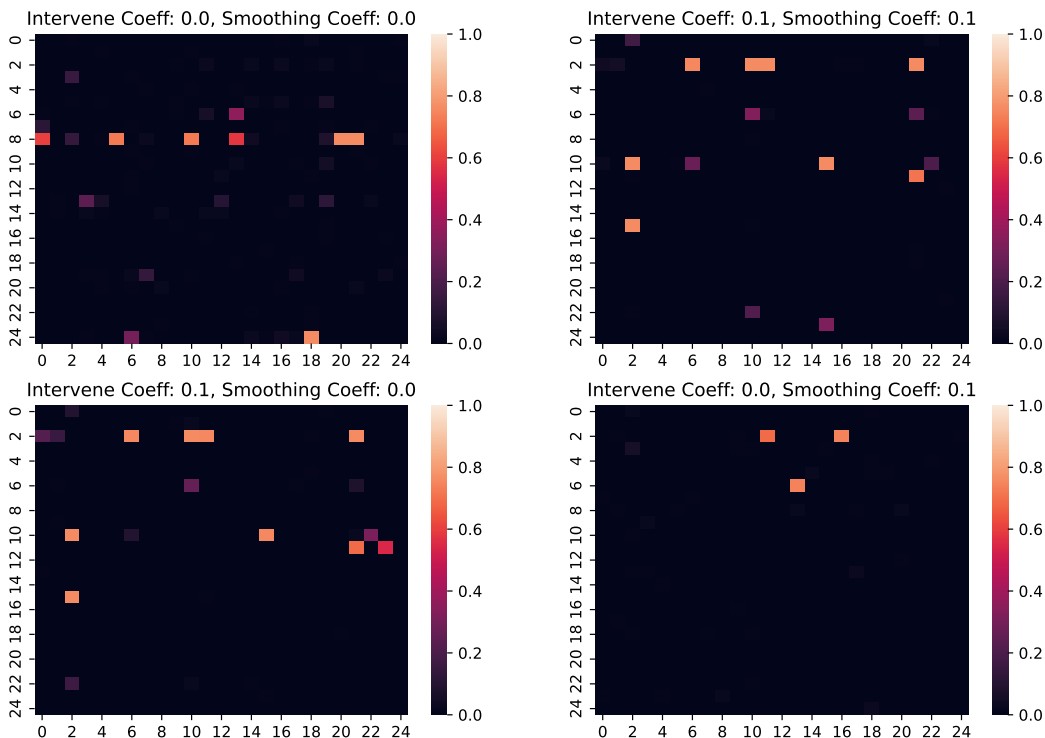

Figure 14: Amortized Networks Interventions in probabilities under different constraints.

## K  ADDITIONAL DETAILS OF TRAFFIC DATA EXPERIEMENT

### K.1  DATASET SETUP

To simulate real-world traffic, based on the road types shown in Fig. 6, we design a road network with four types: intersections with one or two lanes, and T-Junction with one or two lanes. Specifically, we let the speed of the road be 8m/s or 11m/s randomly. Then, we generate car trips by the random generation tool from the SUMO package. We make such a simulation for 1000s at one run. After this playing, we can get the simulation results including emissions (e.g. $CO_2$, $CO$), positions, speed, and lane id (i.e. the car runs in which lane with lanes more than one in a road) of each car at each time step. Given the generated summary data from SUMO, we then count the congestion event (i.e. the car speed less than 0.5m/s) for the following analysis.

### K.2  ADDITIONAL RESULTS AND FIGURE

We show the intensity cost of the learning process for four types of roads in our simulation setup in Fig. 15. For both our model (meta) and our model (train from scratch), the cost trend will converge after several time steps, which proves that our model has great learning and generalization ability.

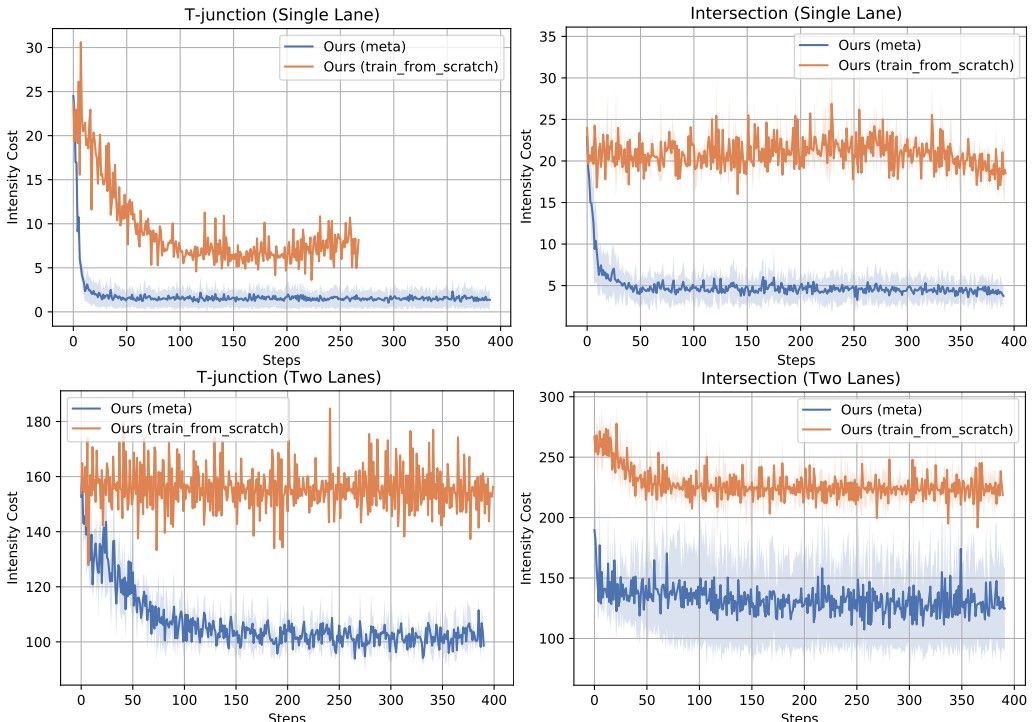

Figure 15: Intensity cost of four types road.

### K.3 CASE STUDY

We further provide a case study to show the great interpretation capacity of our model. In the T-junction with single lane scenario, there are 2 discrete actions, corresponding to the following green phase configurations in Fig. 16. The traffic light is marked as node 10, while the other three lanes are marked as node 8, node 11, and node 12 in our Sumo simulation setup.

Real-world traffic can be represented as an NJODE model corresponding with the change of traffic lights. In our simulation, when node 10 (i.e. traffic light) becomes green, the lane controlled by this traffic signal will connect, which is represented as 1 in Fig. 17; otherwise, the connection between two lanes is disconnected and is represented as 0. For instance, in the first sub-graph of Fig. 17, the car can move from node 11 to node 12 under the control of its traffic signal. The learned policy of our model is exactly consistent with real traffic trips, which demonstrates that our model has great adaptation ability in uncovering real-world network interventions.

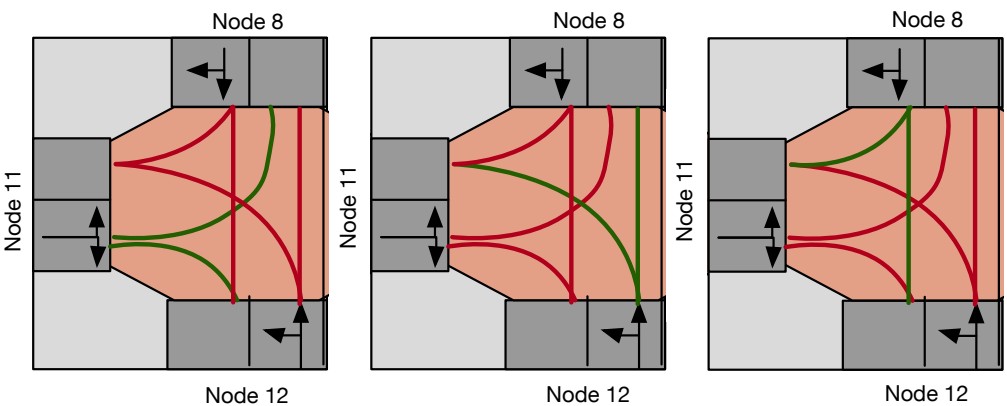

Figure 16: Discrete actions of the T-junction (single lane).

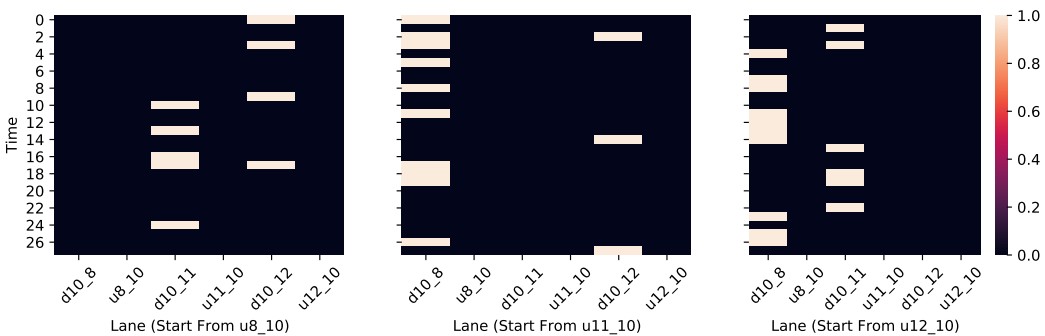

Figure 17: The learned policy of traffic generated from our model.

