### E.1  THEORETICAL JUSTIFICATION

Given event sequences $\mathcal{H}^{t-} := \{(t_i, n_i)\}_{t_i < t}$, we use $\mathbf{x}^t = \{x_1^t, \ldots, x_N^t\}$ to record the incremental number of events for different nodes starting from within $[t + \Delta t)$. We consider a stochastic dynamic system based on the nonlinear system (2-4) presented in the manuscript:

$$\begin{cases} \mathbf{h}_n^{t_0} = \mathbf{h}_n^0, \\ \frac{d\mathbf{h}_n^t}{dt} = f(\mathbf{h}_n^t), \\ \lim_{\epsilon \to 0} \quad \mathbf{h}^{t_i+\epsilon} = \sum_{m \in \mathcal{N}_n} w_{m \to n} \cdot \phi(\mathbf{h}_m^{t_i}, x_m^{t_i}), \end{cases} \tag{12}$$

where $f : \mathbb{R}^d \to \mathbb{R}^d$ and $\phi : \mathbb{R}^d \to \mathbb{R}$ are two Lipschitz functions. The emission/intensity function for the dynamic is defined as $\Lambda(t|\mathbf{h}^{t-}) := g_\Lambda(\mathbf{h}^{t-} \cdot \mathbf{w}) = g_{\Lambda,\mathbf{w}}(\mathbf{h}^{t-})$ where $g_{\Lambda,\mathbf{w}} : \mathbb{R}^{N \times d} \to \mathbb{R}^N$ is a Lipschitz activation function, $\mathbf{w} \in \mathbb{R}^d$ is a linear layer, and $\mathbf{h}^{t-} \in \mathbb{R}^{N \times d}$ is the left continuous points before jump. We use $\lambda_m^t$ to denote $m$-th element of $\Lambda(t)$ and thus $\mathbf{x}_m^{t_i} \sim \mathrm{Poi}(\lambda_m(t_i|\mathbf{h}_n^{t_i-}))$ is the number of incremental events in $[t_i, t_i + \Delta t)$ for node $m$. The ground truth cumulative cost $J(t)$

at finite horizon time $t$ is given by the Monte Carlo Estimator, i.e.,

$$J(t) := \sum_{n=1}^{N} \sum_{i=1}^{t} \mathbb{E}[x_n^i]. \tag{13}$$

Instead of extensively performing Monte Carlo simulation to obtain the cost estimator, we introduce a **_Mean Field Estimator_** $\hat{J}(t)$ for $J(t)$ by averaging out the interaction effect by $x_m^{t_i}$ in the dynamic system (12), i.e.,

$$\hat{J}(t) := \sum_{n=1}^{N} \sum_{i=1}^{t} \hat{\lambda}_n(i). \tag{14}$$

where $\hat{\lambda}_n(t) := \lambda_n(t|\hat{\mathbf{h}}_n^{t-})$ and $\hat{\mathbf{h}}_n^{t-}$ is the mean field estimator for $\mathbf{h}_n^{t-}$ and is derived from a deterministic dynamic by replacing the stochastic discrete update term in system (12) with $\phi(\mathbf{h}_m^{t_i}, \hat{\lambda}_m^{t_i})$. We let the starting point of $\hat{\mathbf{h}}_n^{t-}$ be the same point as $\mathbf{h}_n^{t-}$, i.e., $\hat{\mathbf{h}}_n^0 = \mathbf{h}_n^0, \forall n \in \{1, 2, \cdots, N\}$.

**Proposition 1.** *Given $f$ and $g$ are Lipschitz in Sys. (12), let $\mathbf{h}_n^{t_i^-}$ be the left continuous point of $\mathbf{h}_n^{t_i}$ in the Sys. (12), then it can be recursively expressed by a **Lipschitz** operator $\mathcal{T}_n : \mathbb{R}^{N \times d} \times \mathbb{R}^N \to \mathbb{R}^d$ :*

$$\mathbf{h}_n^{t_i^-} = \mathcal{T}_n(\mathbf{h}^{t_{i-1}^-}, \mathbf{x}^{t_{i-1}}) \tag{15}$$

*where $\mathbf{h}^{t_{i-1}^-} \in \mathbb{R}^{N \times d}$, and $\mathbf{x}^{t_{i-1}} \in \mathbb{R}^N$. More importantly, let $\lambda_{max}$ be the maximum spectrum of influence matrix $\mathbf{W}$, when $L_f$, $L_\phi$, and $\lambda_w$ are smaller than 1, we have $L_{\mathcal{T}_n} < 1$.*

*Proof.* We define $\Phi(\mathbf{h}^{t_i}, \mathbf{x}^{t_i}) := [\phi(\mathbf{h}_1^{t_i}, \mathbf{x}_1^{t_i}), \phi(\mathbf{h}_2^{t_i}, \mathbf{x}_2^{t_i}), \cdots, \phi(\mathbf{h}_N^{t_i}, \mathbf{x}_N^{t_i})]^T \in \mathbb{R}^{N \times d}$ as the discrete kernel in System (12), we can represent $\mathbf{h}_n^{t_i^-}$ as:

$$\mathbf{h}_n^{t_i^-} = \mathcal{T}_n(\mathbf{h}^{t_{i-1}^-}, \mathbf{x}^{t_{i-1}}) = \mathbf{w}_n^T \Phi(\mathbf{h}^{t_{i-1}^-}, \mathbf{x}^{t_{i-1}}) + \int_{t_{i-1}^+}^{t_i^-} f(\mathbf{h}_n^s) \, ds,$$

where $\mathbf{w}^T$ is the $n$-th row of the influence matrix $\mathbf{W}$. Since $f$ and $\phi$ are Lipschitz and the composition of Lipschitz functions is also Lipschitz, so $\mathcal{T}_n$ is Lipschitz. Since $L_{\mathcal{T}} \leq \lambda_{max} \cdot (L_f)^m \cdot (L_\phi)^N$ where $m$ is the number of summation in the intergral term, we have $L_{\mathcal{T}_n} < 1$ when $L_f$, $L_\phi$, and $\lambda_w$ are smaller than 1. $\square$

**Proposition 2.** *Given $f$ and $g$ are Lipschitz in Sys. (12), let $\hat{\mathbf{h}}_n^{t_i^-}$ be the left continuous point of $\hat{\mathbf{h}}_n^{t_i}$ in the deterministic version of the presented Sys. (12) by replacing $x_m^{t_i}$ with $\hat{\lambda}_m(t_i)$, then it can be recursively expressed by a **Lipschitz** operator $\mathcal{T}_n : \mathbb{R}^{N \times d} \times \mathbb{R}^N \to \mathbb{R}^d$ :*

$$\hat{\mathbf{h}}_n^{t_i^-} = \mathcal{T}_n(\mathbf{h}^{t_{i-1}^-}, \hat{\Lambda}(t_{i-1})) \tag{16}$$

*where $\hat{\Lambda}(t_{i-1}) = [\hat{\lambda}_1(t_{i-1}), \hat{\lambda}_2(t_{i-1}), \cdots, \hat{\lambda}_N(t_{i-1})]$.*

*Proof.* This is a direct result from Prop. 1. $\square$

**Lemma 1.** *Let $\hat{\mathbf{h}}_n^i$ be the mean field estimator for $\mathbf{h}_n^i$, suppose $f$ and $\phi$ are Lipshitz , and $\forall n, t, \lambda_n(t)$ is bounded by $K$, we have:*

$$\mathbb{E}_{\mathbf{h}_n^{i-}} \left[ ||\mathbf{h}_n^{i-} - \hat{\mathbf{h}}_n^{i-}|| \right] \leq L_{\mathcal{T}_n} \mathbb{E}_{\mathbf{h}_n^{i-1-}} \left[ ||\mathbf{h}_n^{i-1-} - \hat{\mathbf{h}}_n^{i-1-}|| \right] + M_n, \tag{17}$$

*where $M_n = L_{\mathcal{T}_n}(K^{1/2} + K)$. Moreover, when $L_f < 1$ and $L_\phi < 1$, we have:*

$$\mathbb{E} \left[ ||\mathbf{h}_n^i - \hat{\mathbf{h}}_n^i|| \right] \leq \left( 1 - (L_{\mathcal{T}_n})^i \right) \cdot \frac{M_n}{1 - L_{\mathcal{T}_n}}. \tag{18}$$

*Proof.* For simplicity, we use $\mathbf{h}_n^i$ to denote $\mathbf{h}_n^{i-}$ and $\hat{\mathbf{h}}_n^i$ to denote $\hat{\mathbf{h}}_n^{i-}$. Here we prove the first inequality: By Prop. 1 and Prop. 2, we can decompose $\mathbf{h}_n^i$ and $\hat{\mathbf{h}}_n^i$, i.e.,

$$\mathbb{E}_{\mathbf{h}_n^i}\Big[||\mathbf{h}_n^i - \hat{\mathbf{h}}_n^i||\Big] = \mathbb{E}_{\mathbf{h}_n^{i-1},\mathbf{x}^{i-1}}\Big[||\mathcal{T}_n(\mathbf{h}^{i-1},\mathbf{x}^{i-1}) - \mathcal{T}_n(\hat{\mathbf{h}}^{i-1},\hat{\Lambda}^{i-1})||\Big] \tag{19}$$

$$\leq \mathbb{E}_{\mathbf{h}_n^{i-1},\mathbf{x}^{i-1}}\Big[L_{\mathcal{T}_n}||(\mathbf{h}^{i-1},\mathbf{x}^{i-1}) - (\hat{\mathbf{h}}^{i-1},\hat{\Lambda}^{i-1})||\Big] \tag{20}$$

$$= \mathbb{E}_{\mathbf{h}_n^{i-1},\mathbf{x}^{i-1}}\Big[L_{\mathcal{T}_n}\sqrt{||\mathbf{h}^{i-1} - \hat{\mathbf{h}}^{i-1}||_{\mathbf{h}}^2 + ||\mathbf{x}^{i-1} - \hat{\Lambda}^{i-1}||_{\mathbf{x}}^2}\Big] \tag{21}$$

$$\leq \mathbb{E}_{\mathbf{h}_n^{i-1},\mathbf{x}^{i-1}}\Big[L_{\mathcal{T}_n}\big[||\mathbf{h}^{i-1} - \hat{\mathbf{h}}^{i-1}||_{\mathbf{h}} + ||\mathbf{x}^{i-1} - \hat{\Lambda}^{i-1}||_{\mathbf{x}}\big]\Big] \tag{22}$$

$$= \mathbb{E}_{\mathbf{h}_n^{i-1},\mathbf{x}^{i-1}}\Big[L_{\mathcal{T}_n}\big[||\mathbf{h}^{i-1} - \hat{\mathbf{h}}^{i-1}||_{\mathbf{h}} + ||\mathbf{x}^{i-1} - \Lambda^{i-1} + \Lambda^{i-1} - \hat{\Lambda}^{i-1}||_{\mathbf{x}}\big]\Big] \tag{23}$$

$$\leq L_{\mathcal{T}_n}\mathbb{E}_{\mathbf{h}_n^{i-1}}\Big[||\mathbf{h}^{i-1} - \hat{\mathbf{h}}^{i-1}||\Big] + \mathbb{E}_{\mathbf{x}^{i-1}}\Big[L_{\mathcal{T}_n}||\mathbf{x}^{i-1} - \Lambda^{i-1}||\Big] + L_{\mathcal{T}_n}||\Lambda^{i-1} - \hat{\Lambda}^{i-1}|| \tag{24}$$

$$= L_{\mathcal{T}_n}\mathbb{E}_{\mathbf{h}_n^{i-1}}\Big[||\mathbf{h}^{i-1} - \hat{\mathbf{h}}^{i-1}||\Big] + L_{\mathcal{T}_n}\mathbb{E}_{\mathbf{x}^{i-1}}\Big[||\mathbf{x}^{i-1} - \Lambda^{i-1}||\Big] + L_{\mathcal{T}_n}||\Lambda^{i-1} - \hat{\Lambda}^{i-1}|| \tag{25}$$

$$\leq L_{\mathcal{T}_n}\mathbb{E}_{\mathbf{h}_n^{i-1}}\Big[||\mathbf{h}^{i-1} - \hat{\mathbf{h}}^{i-1}||\Big] + L_{\mathcal{T}_n} \cdot (K^{1/2} + K) \tag{26}$$

$$= L_{\mathcal{T}_n}\mathbb{E}_{\mathbf{h}_n^{i-1}}\Big[||\mathbf{h}^{i-1} - \hat{\mathbf{h}}^{i-1}||\Big] + M_n \tag{27}$$

where the second inequality follows by the Lipschitz property of $\mathcal{T}_n$, the forth inequality follows triangular inequality, and the last inequality follows by Jensen's inequality, i.e., $\bigg(\mathbb{E}_{\mathbf{x}^{i-1}}\Big[||\mathbf{x}^{i-1} - \Lambda^{i-1}||\Big]\bigg)^2 \leq \mathbb{E}_{\mathbf{x}^{i-1}}\Big[||\mathbf{x}^{i-1} - \Lambda^{i-1}||^2\Big] = \Lambda^{i-1} \leq K$ and $||\Lambda^{i-1} - \hat{\Lambda}^{i-1}|| \leq ||\Lambda^{i-1}|| \leq K$ $\quad\square$

*Proof.* Here we prove the second inequality:
From Eq. (17), we can derive its fixed point $x^* := \mathbb{E}_{\mathbf{h}_n^*}\