# OpenReview forum: "Amortized Network Intervention to Steer the Excitatory Point Processes"
_ICLR.cc/2024/Conference — ICLR 2024 poster_

### Official Review · Reviewer_ya7t · 2023-10-31

**Soundness:** 2 fair
**Presentation:** 3 good
**Contribution:** 2 fair
**Rating:** 5
**Confidence:** 3

**Summary:**

The paper proposed a model-based reinforcement learning approach to model networked point processes. Considerations are given on the scalability of the proposed method. The paper describes an amortized policy approach to deal with the scalability issue. The proposed method is applied to synthetic data, real-world covid data, and real-world traffic data.

**Strengths:**

* originality: the problem formulation of networked point processes described by the author has some originality.
* quality: the proposed method makes sense to me. The scalability issue is highlighted in particular. I think the scalability issue is an important one to make the algorithm pratical. The authors proposed the use of amortized policy to deal with this issue.
* presentaiton is clear. I can follow the rationale of the paper.
significance: the proposed method is associated with application scenarios of high significance.

**Weaknesses:**

* I appreciate the authors applying their proposed method to two important pratical problems. However, I think the paper can benefit from comparison to alternative methods.

* Using intensity cost as the evaluation metric is also somewhat obscured.

**Questions:**

* Is it possible for the authors to describe the potential limitation due to the use of mean field approximation for reward modeling?
* Are there experiments to demonstrate the incorporation of fairness constraints and more?
* "For instance, when regulating the coronavirus, government interventions must balance health concerns with economic implications and public sentiment." It is not clear whether the experiments described in the paper achieve such a balance.

---

> ### Author Response · Authors · 2023-11-22
> **Response to reviewer ya7t [Q1-4]**
>
> **[Summary]** We first want to appreciate reviewer ya7t a lot for your acknowledgment of the innovation and contributions of our work including 1) “the problem formulation of networked point processes is original”, 2) “the scalability issue is addressed well by the proposed amortized policy”, 3) “the presentation is clear”, and 4) “the proposed method is associated with application scenarios of high significance”. We then provide point-wise responses below to address the concern.
>
> > **[Q1]**: Alternative methods.
>
> As the suggestion to add more comparisons, we implemented two common RL baselines: SAC and PPO on the COVID data. The empirical results have been incorporated into Appendix I.2 (Figure 13). By using NJODE as our simulator, we observe that the proposed method has successfully reduced the number of infested people (with an average reduced intensity: 0.49) in six counties among the chosen nine counties while SAC and PPO struggle to have a positive intervention effect to control the pandemic.
>
> One interesting alternative method would be to combine causal inference with RL. In fact, in our experiment, we have introduced a pre-learned Boolean mask to determine the support of the action space, which can be interpreted as leveraging the Granger causal structure to guide policy optimization. Conducting online causal inference given the data collected by RL provides an interesting direction, yet is still open in our context. Particularly, ensuring the identifiability of causal effect estimation requires careful investigation, especially considering our dynamics are modeled as neural ODEs with jumps.
>
> A most related reference we found is [1], which asks the counterfactual question given the observed temporal point processes by behavior policy and is related to off-policy evaluation for target policy.
>
> References:
> [1] Noorbakhsh, K., & Rodriguez, M. (2022). Counterfactual temporal point processes. Advances in Neural Information Processing Systems, 35, 24810-24823.
>
> > **[Q2]**: Intensity cost as the evaluation metric.
>
> Here the intensity cost refers to the derived average intensity throughout the whole intervention process for each node. Since our model is based on the Counting process, instead of simulating the process itself, this average intensity will give us a fair and efficient indicator of how fast the counting process will grow during a given period. Moreover, the economic effect caused by the growing number of infected people and the carbon emissions because of the traffic congestion are all highly related to the intensity of newly infected people and newly arrived vehicles. Thus, in this paper, we think it is fair to use intensity cost to evaluate t.
>
> > **[Q3]**: Potential limitation due to the use of mean-field approximation for reward modeling.
>
> For the limitation, we have a detailed error-bound analysis for the presented mean-field approximation which is covered in the updated manuscript Appendix E.2. This analysis is built upon a Lipschitz assumption of the given dynamic. The theoretical analysis further provides the insight that, as long as the spectral norm of the ODE neural layer is smaller than one, the mean-field approximation error can be well bounded.
>
> > **[Q4]**: Experiments to demonstrate the incorporation of fairness constraints.
>
> We added experiments by considering two soft fairness-related constraints and provided an empirical study in Appendix Section I.2. Specifically, two fairness augmentation terms were added to the objective function: 1) intervention budget cost; and 2)  policy smoothing cost. The first cost penalizes implementing the intervention policy that exceeds the predefined budget and the second one trades off the policy-changing frequency. In addition to the two terms under consideration, more realistic constraints relevant to specific scenarios can also be included in practice.

---

> ### Author Response · Authors · 2023-11-22
> **Response to reviewer ya7t [Q5]**
>
> > **[Q5]**: Experiment to show the balance of health concerns with economic implications and public sentiment.
>
> During the intervention, we have three constraints related to the balance of health concerns and economic implications:
> 1. We have set up a hard constraint and implemented a dynamic mask to explicitly exclude actions that fall outside the feasible space, ruling out the action that consecutively locks down one county. This maximum length of consecutive lockdown for one city can give us a trade-off between enforcing long-time continuous lockdowns to control pandemics and allowing temporary ‘freedom’ to recover the economy.
> 2. We introduce a policy smoothing cost to penalize the abrupt difference between consecutive policies. This means a larger weight will heavily discourage the agent from frequently changing the policy (which will incur more financial burden) while a smaller weight will less discourage a frequent policy change.
> 3. We also introduce an intervention budget cost and penalize the intervention policy that exceeds the predefined budget. Intuitively, a large weight of the intervention cost will discourage the agent from intervening more edges and thus have limited control over the pandemic while a small weight of the intervention cost allows more connections to be controlled but will generate a higher expense.
>
> To understand this trade-off, we have performed experiments of choosing different weights of intervention budget cost and policy smoothing cost in Appendix I.2.

---

> ### Author Response · Authors · 2023-11-23
> **Sincerely expecting further discussion from Reviewer ya7t**
>
> Dear Reviewer ya7t,
>
> Thank you very much for your commitment and effort in evaluating our paper. We understand that this is a hectic time, and we are writing to gently remind you about providing your insights on our rebuttal. As the discussion phase is drawing to a close, your feedback would be invaluable to us.
>
> Should you have any further thoughts or recommendations about our work, we are eager to discuss them with you.
>
> We eagerly await your reply.
>
> Thank you once again,
>
> The Authors

---

### Official Review · Reviewer_hvRi · 2023-11-03

**Soundness:** 3 good
**Presentation:** 3 good
**Contribution:** 3 good
**Rating:** 6
**Confidence:** 3

**Summary:**

The paper discusses a method for addressing the challenge of large-scale network intervention in scenarios like controlling the spread of infectious diseases or managing traffic congestion. The approach uses model-based reinforcement learning with neural ODEs to predict how excitatory processes in a network will change over time as the network structure evolves. It incorporates Gradient-Descent based Model Predictive Control (GD-MPC) to provide flexibility in policy development, accommodating prior knowledge and constraints. To handle complex planning problems with high dimensionality, the authors introduce an Amortize Network Interventions (ANI) framework, which allows pooling of optimal policies from historical data and various contexts while ensuring efficient knowledge transfer. This method has broad applications, including disease control and traffic optimization.

**Strengths:**

The paper is well-motivated and the proposed model is technically sound and can notably handle large-scale systems. The overall writing is easy to follow. The experiment section is comprehensive though missing some baselines.

**Weaknesses:**

In the experiment section, why baseline comparison is only limited on one synthetic dataset? Also, can the author explains why the NHPI baseline almost have a constant intensity cost in Figure 1?

**Questions:**

For adding interventions, can we also consider causal-inference-based methods as valid comparison?

---

> ### Author Response · Authors · 2023-11-22
> **Reponse to reviewer hvRi**
>
> **[Summary]** Many thanks to reviewer hvRi for your positive comments and recognition of our contribution including 1) “The paper is well-motivated and technically sound”; 2) “The proposed method can notably handle large-scale systems”; 3) “The overall writing is easy to follow” and 4) “The experiment section is comprehensive”. We would like to address your concerns one by one.
>
> > **[Q1]**: NHPI Baselines and Additional Baselines.
>
> As the suggestion to add more baselines, we implemented two common RL baselines: SAC and PPO on the COVID data. The empirical results have been incorporated into Appendix I.2 (Figure 13). By using NJODE as our simulator, we observe that the proposed method has successfully reduced the number of infested people (with an average reduced intensity: 0.49) in six counties among the chosen nine counties while SAC and PPO struggle to have a positive intervention effect to control the pandemic.
>
> As for the constant intensity cost of the NHPI baseline, it was originally proposed on less dense point processes (i.e., tweet data) and used an event-driven RL to steer the point process. We conjecture the less satisfied performance is mainly caused by two aspects:
> 1. NHPI used a transformer-based sequential model on historical events to learn the dynamic model. This indeed is not easy to capture the inherent connections between different types of events and also not the best model representation for event sequences generated from a dense graph. On the contrary, the proposed NJODE is built on explicit graph embeddings and has a strong potential to model the mutual interaction between different types of events.
> 2. NHPI learns a stationary value function through the Bellman function on an infinite horizon semi-MDP. We think the single-step TD learning for semi-MDP presented in NHPI could cause bias to the learned value function, especially in the non-stationary environment. Instead, our policy is learned from a receding horizon window without learning additional value functions and can be adapted to a new model by iteratively updating the parameters of neural ODE.
>
> > **[Q2]**: Causal-inference-based baseline method
>
> It is a great idea to combine causal inference with RL. In fact, in our experiment, we have introduced a pre-learned Boolean mask to determine the support of the action space, which can be interpreted as leveraging the Granger causal structure to guide policy optimization. Conducting online causal inference given the data collected by RL provides an interesting direction, yet is still open in our context. Particularly, ensuring the identifiability of causal effect estimation requires careful investigation, especially considering our dynamics are modeled as neural ODEs with jumps.
> A most related reference we found is [1], which asks the counterfactual question given the observed temporal point processes by behavior policy and is related to off-policy evaluation for target policy.
>
> References:
> [1] Noorbakhsh, K., & Rodriguez, M. (2022). Counterfactual temporal point processes. Advances in Neural Information Processing Systems, 35, 24810-24823.

---

> ### Author Response · Authors · 2023-11-23
> **Sincerely expecting further discussion from Reviewer hvRi**
>
> Dear Reviewer hvRi,
>
> Thank you very much for your commitment and effort in evaluating our paper. We understand that this is a hectic time, and we are writing to gently remind you about providing your insights on our rebuttal. As the discussion phase is drawing to a close, your feedback would be invaluable to us.
>
> Should you have any further thoughts or recommendations about our work, we are eager to discuss them with you.
>
> We eagerly await your reply.
>
> Thank you once again,
>
> The Authors

---

### Official Review · Reviewer_YqcV · 2023-11-07

**Soundness:** 3 good
**Presentation:** 3 good
**Contribution:** 3 good
**Rating:** 6
**Confidence:** 4

**Summary:**

This paper proposes a re-enforcement learning algorithm to adaptively change the network structure to steer the observed event counts over a potentially large network. The proposed algorithm is carefully crafted to model network point process data with complicated data structures, which is desirable for solving large-scale real-data applications. The proposed algorithm is theoretically sound, and its effectiveness is demonstrated through simulation studies and real data applications.

**Strengths:**

The paper is well written, the presentation is clear and the idea is sound.

**Weaknesses:**

More details are needed in some components of the proposed model, including the mean field approximation for the rewarding model and the construction of the amortized policy.

**Questions:**

1. The review of the work on networked excitatory point processes seems to only focus on work that used ODEs. However, I would like to point out that there are also other streams of research on networked excitatory point processes, for example, [1], [2], [3]. It is better to conduct a more comprehensive review of the topic.

2. In the definition of the temporal graph network, what are the definitions of edges in the two examples given in the paper? I do not find formal definitions. Could you please clarify?

3. On page 3, it is claimed that the "high-dimensional event sequences $\{X_t\}_{t\ge 0}$ has a stationary dynamics". What do you mean by "stationary"? If $X_t$ is the accumulated counts up to time $t$, it is unlikely to be a stationary time series. Please clarify.

4. Is the influence matrix $W$ in equation~(4) a given matrix or learned from the data?

5. The use of mean field approximation on page 4 seems to be rather ad-hoc, without much theoretical or even heuristic justifications. Can you give an example to show the difference between the actual reward and the one calculated based on the mean-field approximation? Are they actually close? Perhaps some simulation studies can be carried out to show the difference in actual reduction in intensity using these two reward objectives in some simple settings.

6. On page 5, it is stated that "Given a sequence of local policies $\{\pi_i\}$, $1\le i\le M$,  addressing $M$ distinct sub-problems, our goal is to create an amortized policy $\pi_{amo}$". However, I did not find any detail on how this goal can be achieved. Can you clarify?

7. The label of the y-axis in Figure 3 reads "Intensity Cost". What is the definition of the "Intensity Cost"?


[1] Delattre, S., Fournier, N., & Hoffmann, M. (2016). Hawkes processes on large networks.

[2] Fang, G., Xu, G., Xu, H., Zhu, X., & Guan, Y. (2023). Group network Hawkes process. Journal of the American Statistical Association, (just-accepted), 1-78.

[3] Cai, B., Zhang, J., & Guan, Y. (2022). Latent Network Structure Learning From High-Dimensional Multivariate Point Processes. Journal of the American Statistical Association, 1-14.

---

> ### Author Response · Authors · 2023-11-22
> **Reponse to reviewer YqcV**
>
> **[Summary]**: We appreciate that reviewer YqcV has a positive impression of the **“sound idea”** and **“well written”** of our paper. To address your questions about the details of our method, we provide point-wise responses as follows.
>
> > **[Q1]**: Related work of networked excitatory point processes.
>
> We have carefully reviewed and incorporated the listed publications in Appendix Section A (related work) in our updated version.
>
> > **[Q2-Q4]**: Definition clarifications of the model details.
>
> Here we make the following clarifications.
> 1. **Edge**: For pandemic control cases, the node is defined as county/city, and the edges are defined as the accessibility from one city to another city (see Figure 1). For traffic control cases, the node is defined as a road lane, and edges are defined as the connections between different lanes.
> 2. **Stationary Dynamic**: Here $X_{t}$ refers to the number of a sequence of spike counts within a time step, so $X_{t}$ can be stationary in a long-term period. Moreover, under a careful choice of the initial parameters for the neural network (as the contraction condition derived in Appendix E), the dynamic system is also stationary.
> 3. **Influence matrix $W$**: In our cases, only the support of $W$ matters, and it can be obtained from prior knowledge. The reason is if we jointly learn the influence matrix $W$ and the discrete kernel function $\phi$ in the proposed system (Eq. (4) in the manuscript), W will become not identifiable and lose its original physical meaning. Therefore, we choose to fix the support of $W$ (i.e., $w_{ij} \in [0,1] $) and the specific value can be inferred from the data-driven method. For instance, in the COVID case, the influence matrix is restricted to $[0,1]$ and is proportional to the inverse distance between different counties.
>
> We have also updated the above clarifications according to the updated manuscript.
>
> > **[Q5]**: Mean-field approximation justification
>
> Here, we provide a brief justification for the mean-field approximation. For supplementary numerical and theoretical justification, please refer to our modified draft (Appendix E).
>
> To evaluate the policy gradient at each iteration, one needs to get sufficient roll-out trajectories. Numerically, obtaining one roll-out trajectory requires solving ODEs repeatedly within each time interval, and sampling random counts at the end of each interval, which will be fed back to the system to update the state. The mean-field approximation, however, only needs to compute the roll-out once – it uses a deterministic trajectory to approximate the mean of the stochastic trajectories, by replacing the random counts sampled at the end of each interval by their means.
>
> Mean-field approximation dramatically reduces the computational burden in policy optimization. We added numerical results (shown in Fig. 11, Appendix E.2) and theoretical analysis (in Appendix E.1) to demonstrate the approximation accuracy.
>
> The theoretical analysis further provides the insight that, as long as the spectral norm of the ODE neural layer is smaller than one, the mean-field approximation error can be well bounded.
>
> > **[Q6]**: Amortized policy construction
>
> Mathematically, the proposed amortized policy $\pi_{ani}$ is learned over a distribution of random tasks from $\mathcal{M}$, i.e., $\pi_{ani} = E_{M_{i}}[\pi_{i}]$ where $M_{i} \sim \mathcal{M}$. The meta-policy gradient over different tasks is then updated in a similar fashion with Reptile [1]. We have clarified the point in the updated version.
>
> [1] Nichol, A., Achiam, J., & Schulman, J. (2018). On first-order meta-learning algorithms. arXiv preprint arXiv:1803.02999.
>
>
> > **[Q7]**: Figure label clarification
>
> For brevity, here intensity cost means the average intensity throughout a fixed period and over all the nodes within a local region. We use this indicator to reflect how fast the pandemic/traffic will grow within a multivariate system. We have clarified this point in the updated version.

---

> ### Author Response · Authors · 2023-11-23
> **Sincerely expecting further discussion from Reviewer YqcV**
>
> Dear Reviewer YqcV,
>
> Thank you very much for your commitment and effort in evaluating our paper. We understand that this is a hectic time, and we are writing to gently remind you about providing your insights on our rebuttal. As the discussion phase is drawing to a close, your feedback would be invaluable to us.
>
> Should you have any further thoughts or recommendations about our work, we are eager to discuss them with you.
>
> We eagerly await your reply.
>
> Thank you once again,
>
> The Authors

---

### Official Review · Reviewer_9miN · 2023-11-09

**Soundness:** 3 good
**Presentation:** 2 fair
**Contribution:** 2 fair
**Rating:** 5
**Confidence:** 2

**Summary:**

The paper looks at both the learning and optimal intervention planning problem for a large-scale network whose evolutions include both continuous state drift and discrete jumps. The authors propose to learn the transition of the states via MLE. Directly dealing with the large-scale problem is intractable. Hence, the authors also propose the Amortized Policy Learning framework to
1) segment the network into pieces and each time pick one subnetwork to update the model estimation;
2) update the policy parameters with data collected from the current subnetwork using common feature representation learned by minimizing the bi-contrastive loss.

The authors apply the proposed framework to tasks including synthetic data, covid data and traffic data where improvements are obtained and show that the proposed policy equivalent embeddings successfully decouple the position embeddings and value embeddings.

**Strengths:**

The authors' contributions include:
1) Formulate the problem as a RL problem.
2) Propose to decompose the problem into subproblems of smaller scale and learn a policy that can generalize to the full-scale problem via   ensuring permutation equivalence.
3) Test and compare the methods to previous methods on several settings of practical interest. Also, the authors show concrete evidence of the out-of-distribution generalization power of the proposed method.

**Weaknesses:**

In my point of view, the authors did not clarify their contributions.

In terms of problem formulation, modeling the discrete events by counting the occurrences in a time window should not be counted as the novelty. In terms of model learning, the authors are basically using the MLE method, which is standard in model-based RL.
Policy learning within the permutation equivalence class through learning embedding $p^t, m^t$ via contrastive method is an interesting idea. However, the authors do not differentiate their work from (Chen et al., 2020b).

More crucially, each part in a large networks may have both global and local patterns. How to deal with the local patterns in model learning and policy optimization is not clear. Or the authors are just trying to obtain a policy that only has "overall" good performance.

Another question is how to deal with the soft/hard constraints. I did not find evidence supporting the effectiveness of the constraint ensuring methods.

**Questions:**

As I have mentioned before, is it possible for the current framework to capture the local patterns in a large network and perhaps steer the learned policy towards local patterns while maintaining the generalization power (such as regularized policy optimization).

---

> ### Author Response · Authors · 2023-11-22
> **Response to reviewer 9miN**
>
> **[Summary]**: We first thank reviewer 9miN for the insightful comments, especially for the questions about our high-level contributions and experiment designs, which benefit us to further clarify our paper. We would like to address the concerns one by one.
>
> > **[Q1]**: Contributions of the paper and the difference between our proposed loss and SimCLR.
>
> We highlight the contributions part in the Introduction of the revised manuscript. Our contributions include three aspects: 1) a technically sound method to tackle the challenge of large-scale network intervention, 2) the proposal of a permutation-equivalent intervention paradigm, and 3) comprehensive experiments on traffic congestion and COVID data. Most importantly, our key contribution lies in introducing the permutation-equivalent intervention paradigm within a dynamically evolving large network topology, facilitated by the utilization of the proposed bi-contrastive loss function.
>
> As for the difference between the proposed Bi-contrastive loss and the previous SimCLR (Chen et al., 2020b):
>  - Firstly, unlike SimCLR which directly performs augmentations on visual representations, we augment our network on its latent embedding space by utilizing graph augmentation techniques.
> - Secondly, our model does not require sampling additional data as negative examples and projecting the augmented sample to one embedding. Instead, we only use one anchor sample (here is the embedded network) project the sample to two different embeddings (p and m), and augment the anchor sample by permuting its node orders or adding noise to the node embeddings such that it can form two negative pairs naturally (as depicted in Figure 1). More details will be provided in Appendix Section F in the updated version.
>
> > **[Q2]**: Global and local pattern for network work intervention.
>
> In our model, we have the permutation equivalent property to extract the global invariant patterns. Regarding the local pattern, we use the node embeddings $H_{t}$ learned by the Networked Jump ODE model to store the information. For instance, the elements of node embedding can have the meaning of a local region’s population and economy.  When performing policy optimization, the node embeddings will be used to learn the policy and global invariant patterns. Therefore, our policy can capture both local patterns and global patterns elegantly.
>
> > **[Q3]**: Dealing with soft and hard constraints.
>
> For hard constraints like strictly controlling the maximum number of consecutive interventions, we use a ‘’dynamic mask” in our policy learning. For soft constraints, we can treat the constraints as ‘’regularization/augmentation terms” and add these terms to the objective function.
>
> We added experiments by considering two soft fairness-related constraints and provided an empirical study in Appendix Section I.2. Specifically, two fairness augmentation terms were added to the objective function: 1) intervention budget cost; and 2) policy smoothing cost. The first cost penalizes implementing the intervention policy that exceeds the predefined budget and the second one trades off the policy-changing frequency. In addition to the two terms under consideration, more realistic constraints relevant to specific scenarios can also be included in practice.

---

> ### Author Response · Authors · 2023-11-23
> **Sincerely expecting further discussion from Reviewer 9miN**
>
> Dear Reviewer 9miN,
>
> Thank you very much for your commitment and effort in evaluating our paper. We understand that this is a hectic time, and we are writing to gently remind you about providing your insights on our rebuttal. As the discussion phase is drawing to a close, your feedback would be invaluable to us.
>
> Should you have any further thoughts or recommendations about our work, we are eager to discuss them with you.
>
> We eagerly await your reply.
>
> Thank you once again,
>
> The Authors

---

### Author Response · Authors · 2023-11-22
**Invitation for review feedback**

Dear Esteemed reviewers,

We are grateful to all the reviewers for generously dedicating their time and effort to evaluating our paper. Their constructive feedback and valuable suggestions are helpful to
further enhance the quality of our work. We have incorporated all the recommended changes into the revised manuscript, clearly denoted in "blue".

Thank you for your considerable contributions during the review process. We anticipate further discussions in the future.
Once again, we are grateful for your valuable insights.

Warm wishes,

The authors

---

### Author Response · Authors · 2023-11-23
**General Response**

In addition to the pointwise responses below, here we summarize our major updates, and the edited parts are also highlighted in the manuscript in blue.

**[Q1]: The reason for applying the mean-field approximation for the cumulative cost (@reviewer YqcV, ya7t)**

Here, we provide a brief justification for the mean-field approximation. For supplementary numerical and theoretical justification, please refer to our modified draft (Appendix E).

To evaluate the policy gradient at each iteration, one needs to get sufficient roll-out trajectories. Numerically, obtaining one roll-out trajectory requires solving ODEs repeatedly within each time interval, and sampling random counts at the end of each interval, which will be fed back to the system to update the state. The mean-field approximation, however, only needs to compute the roll-out once – it uses a deterministic trajectory to approximate the mean of the stochastic trajectories, by replacing the random counts sampled at the end of each interval by their means.

Mean-field approximation dramatically reduces the computational burden in policy optimization. We added numerical results (shown in Fig. 11, Appendix E.2) and theoretical analysis (in Appendix E.1) to demonstrate the approximation accuracy.

The theoretical analysis further provides the insight that, as long as the spectral norm of the ODE neural layer is smaller than one, the mean-field approximation error can be well bounded.


**[Q2]: The suggestion to add empirical results by considering fairness constraints (@reviewer 9miN, ya7t)**

As suggested by the reviewer,  we added experiments by incorporating two fairness-related augmentation terms to the objective. For more detail, please refer to Appendix Section H.2 of our modified draft.

| Fairness / Parameters           | $\lambda_{1}=0.0$ | $\lambda_{1}=0.0$ | $\lambda_{1}=0.1$ | $\lambda_{1}=0.1$ |
|---------------------------------|-------------------|-------------------|-------------------|-------------------|
|                                 | $\lambda_{2}=0.0$ | $\lambda_{2}=0.1$ | $\lambda_{2}=0.0$ | $\lambda_{2}=0.1$ |
| Average Reduced intensity       | 0.25(0.06)        | 0.10(0.04)        | 0.06(0.05)        | 0.01(0.03)        |
| Average Lockdown probability (%)| 1.31(0.53)        | 0.43(0.12)        | 1.29(0.27)        | 1.29(0.17)        |

Specifically, two fairness augmentation terms were added to the objective function: 1) intervention budget cost (weighted by $\lambda_{1}$); and 2) policy smoothing cost (weighted by $\lambda_{2}$). The first cost penalizes implementing the intervention policy that exceeds the predefined budget and the second one trades off the policy-changing frequency. The results suggest enforcing smoothing constraints ($i.e., \lambda_{1}=0.0, \lambda_{2}=0.1$) gives us a fairer policy, i.e., less effort to intervene (average lockdown probability: 0.43 \%) but achieve positive control effect (average reduced intensity: 0.10)

In addition to the two terms under consideration, more realistic constraints relevant to specific scenarios can also be included in practice.

**[Q3]: Alternative Methods and Additional Baselines on Covid (@reviewer hvRi, ya7t)**


As the suggestion to consider causal-inference-based methods:

It is a great idea to combine causal inference with RL. In fact, in our experiment, we have introduced a pre-learned Boolean mask to determine the support of the action space, which can be interpreted as leveraging the Granger causal structure to guide policy optimization. Conducting online causal inference given the data collected by RL provides an interesting direction, yet is still open in our context. Particularly, ensuring the identifiability of causal effect estimation requires careful investigation, especially considering our dynamics are modeled as neural ODEs with jumps.
A most related reference we found is [1], which asks the counterfactual question given the observed temporal point processes by behavior policy and is related to off-policy evaluation for target policy.

As for adding more baselines:

In addition, we also implement two common RL baselines: SAC and PPO on the COVID data. The empirical results have been incorporated into Appendix H.2 (Figure 13). By using NJODE as our simulator, we observe that the proposed method has successfully reduced the number of infested people (with an average reduced intensity: 0.49) in six counties among the chosen nine counties while SAC and PPO struggle to have a positive intervention effect to control the pandemic.

References:

[1] Noorbakhsh, K., & Rodriguez, M. (2022). Counterfactual temporal point processes. Advances in Neural Information Processing Systems, 35, 24810-24823.

---

### Meta-Review · Area_Chair_KXua · 2023-12-06

**Metareview:**

The paper considers the problem of networked point processes under a large-scale network whose evolutions include both continuous state drift and discrete jumps. The authors formulate the problem as an RL problem, propose to learn the state transition with ODEs and scale up with Amortize Network Interventions (ANI) framework. Experiments on synthetic traffic congestion data and real-world COVID datasets showed the effectiveness of the solution.  The reviewers appreciate the motivation and the novel idea. There were several concerns including requesting additional comparisons in experiments, clarifying contribution and the usage of the mean-field approximation. During rebuttal period, the authors provided additional experiments (baselines, fairness constraints) and justification. In my opinion, the rebuttal answered most of the questions but the justification for global and local pattern for network work intervention (by Reviewer 9miN) should be further clarified. Overall, I recommend acceptance of the paper.

**Justification For Why Not Higher Score:**

The justification for global and local pattern for network work intervention (by Reviewer 9miN) should be further clarified.

**Justification For Why Not Lower Score:**

The reviewers recognized the importance of the problem and the idea being interesting. The rebuttal and new experimental results answered most of the concerns in reviews.

---

### Decision · Program_Chairs · 2024-01-16

Accept (poster)